# Ubiquitin ligase and signalling hub MYCBP2 is required for efficient EPHB2 tyrosine kinase receptor function

Chao Chang[1,2], Sara L Banerjee[1,3], Sung Soon Park[1,2], Xiao Lei Zhang[1], David Cotnoir-White[1†], Karla J Opperman[4], Muriel Desbois[4,5], Brock Grill[4,6,7], Artur Kania[1,2,3,8]*

[1]Institut de recherches cliniques de Montréal (IRCM), Montréal, Canada; [2]Integrated Program in Neuroscience, McGill University, Montréal, Canada; [3]Division of Experimental Medicine, McGill University, Montréal, Canada; [4]Center for Integrative Brain Research, Seattle Children's Research Institute, Seattle, United States; [5]School of Life Sciences, Keele University, Keele, United Kingdom; [6]Department of Pediatrics, University of Washington School of Medicine, Seattle, United States; [7]Department of Pharmacology, University of Washington School of Medicine, Seattle, United States; [8]Department of Anatomy and Cell Biology, McGill University, Montréal, Canada

*For correspondence:
artur.kania@ircm.qc.ca

Present address: †Modulari-T Biosciences, Montréal, Canada

Competing interest: The authors declare that no competing interests exist.

**Abstract** Eph receptor tyrosine kinases participate in a variety of normal and pathogenic processes during development and throughout adulthood. This versatility is likely facilitated by the ability of Eph receptors to signal through diverse cellular signalling pathways: primarily by controlling cytoskeletal dynamics, but also by regulating cellular growth, proliferation, and survival. Despite many proteins linked to these signalling pathways interacting with Eph receptors, the specific mechanisms behind such links and their coordination remain to be elucidated. In a proteomics screen for novel EPHB2 multi-effector proteins, we identified human MYC binding protein 2 (MYCBP2 or PAM or Phr1). MYCBP2 is a large signalling hub involved in diverse processes such as neuronal connectivity, synaptic growth, cell division, neuronal survival, and protein ubiquitination. Our biochemical experiments demonstrate that the formation of a complex containing EPHB2 and MYCBP2 is facilitated by FBXO45, a protein known to select substrates for MYCBP2 ubiquitin ligase activity. Formation of the MYCBP2-EPHB2 complex does not require EPHB2 tyrosine kinase activity and is destabilised by binding of ephrin-B ligands, suggesting that the MYCBP2-EPHB2 association is a prelude to EPHB2 signalling. Paradoxically, the loss of MYCBP2 results in increased ubiquitination of EPHB2 and a decrease of its protein levels suggesting that MYCBP2 stabilises EPHB2. Commensurate with this effect, our cellular experiments reveal that MYCBP2 is essential for efficient EPHB2 signalling responses in cell lines and primary neurons. Finally, our genetic studies in *Caenorhabditis elegans* provide in vivo evidence that the ephrin receptor VAB-1 displays genetic interactions with known MYCBP2 binding proteins. Together, our results align with the similarity of neurodevelopmental phenotypes caused by MYCBP2 and EPHB2 loss of function, and couple EPHB2 to a signalling effector that controls diverse cellular functions.

## eLife assessment

This **valuable** study identifies an Ephrin type-B Receptor 2 (EPHB2) interactor, MYCBP2, as a potential regulator of EPHB2 stability and function. In contrast to expectations, based on MYCBP2 function in the ubiquitin pathway, loss of function of MYCBP2 resulted in less EPHB2 receptor and

defective EPHB2 function. The paper is supported by a largely **convincing** set of biochemical, cell culture and in vivo experiments.

## Introduction

Eph receptor tyrosine kinases and their membrane-tethered ligands, the ephrins, elicit short distance cell-cell signals that regulate many biological processes. Ephrin-Eph signalling primarily impacts the cytoskeleton with the immobilization of highly dynamic axonal growth cones being a classic example. Other processes that involve changes in transcription, growth, and survival such as angiogenesis, synaptic plasticity, stem cell fate, tumorigenesis, and neurodegeneration also involve the Eph/ephrin system. Many proteins are postulated to couple Eph receptors to different intracellular effectors, but the molecular logic of this diversity remains fragmented (*Kania and Klein, 2016*; *Bush, 2022*).

EphB subfamily members preferentially bind transmembrane ephrin-Bs and although both molecules participate in bidirectional signalling, ephrin-B activation of EphB signalling cascades is more thoroughly studied (*Gale et al., 1996*; *Mellitzer et al., 1999*). To elicit robust Eph receptor forward signalling, ephrins multimerise in signalling clusters by intercalating with Ephs on a signal-recipient cell with array size correlating with signal amplitude (*Kullander et al., 2001*; *Schaupp et al., 2014*). Signalling initiation involves the activation of the receptor tyrosine kinase and phosphorylation of tyrosines proximal to the EphB transmembrane domain (*Soskis et al., 2012*; *Binns et al., 2000*). Eph-evoked cytoskeletal effects such as cell contraction and growth cone collapse result from changes in small GTPase activity modulated by EphB Guanine nucleotide exchange factors (GEFs) and GTPase activating proteins (GAPs) (*Margolis et al., 2010*; *Shi et al., 2007*). Eph signalling has also been linked to fundamental cellular pathways such as the Ras-MAPK pathway, mTOR-regulated protein synthesis, cell division, and survival (*Bush and Soriano, 2010*; *Nie et al., 2010*; *Fawal et al., 2018*; *Genander et al., 2009*; *Depaepe et al., 2005*). Eph forward signalling eventually leads to their internalisation and either recycling or degradation via endosome/lysosome and ubiquitination/proteasome pathways (*Zimmer et al., 2003*; *Okumura et al., 2017*). While identification of a growing number of proteins interacting with Eph receptors has moved the field forward, we have yet to clarify the question of how Eph receptors activate various fundamental cellular processes, often within the same cell type.

Myc-binding protein 2 (MYCBP2), also known as Protein Associated with Myc (PAM) and Highwire, RPM-1, or Phr1 in different species, is a large signalling hub that regulates cytoskeletal dynamics, neuronal development, and axonal degeneration (*Guo et al., 1998*; *Grill et al., 2016*; *Virdee, 2022*). It has an atypical RING ubiquitin ligase activity that inhibits the p38/MAP and JNK kinase pathways thereby regulating cytoskeletal dynamics underlying axonal development and synaptic growth (*Nakata et al., 2005*; *Collins et al., 2006*; *Pao et al., 2018*; *Wan et al., 2000*; *Lewcock et al., 2007*; *Borgen et al., 2017*). MYCBP2 further regulates the Tuberin Sclerosis Complex linked to cell growth (*Han et al., 2012*), initiation of autophagy via ULK (*Crawley et al., 2019*) and NMNAT2-regulated neuronal survival and axonal degeneration (*Babetto et al., 2013*; *Xiong et al., 2012*). Biochemical mapping has shown that human MYCBP2 and its *Caenorhabditis elegans* ortholog RPM-1 rely upon the FBD1 domain to bind the F-box protein FBXO45 that acts as a ubiquitination substrate selector (*Desbois et al., 2018*; *Sharma et al., 2014*). Intriguingly, the neurodevelopmental phenotypes caused by MYCBP2 and EPHB2 loss of function are similar, raising the possibility that these two molecules could function in the same pathway (*Henkemeyer et al., 1996*; *Lewcock et al., 2007*; *Dalva et al., 2000*). Nonetheless, a biochemical or genetic interaction between MYCBP2 and EPHB2 has not been demonstrated in any system.

To shed light on how EphB receptors fulfil their multitude of functions, we used mass spectrometry (MS)-based proteomics to identify multi-effector proteins that bind EPHB2. One of our proteomic hits was MYCBP2, which we demonstrated forms a complex with EPHB2 using a combination of biochemical and cellular assays. Furthermore, we show that this interaction is required for efficient EPHB2 signalling in cell lines and primary neurons. Consistent with these findings, we observed in vivo genetic interactions in *C. elegans* between the Eph receptor, VAB-1, and known RPM-1 binding proteins. Our collective results indicate that the relationship between EPHB2 and MYCBP2 does not appear to involve the ubiquitin ligase activity of MYCBP2, and raise the possibility that MYCBP2 links EPHB2 to diverse fundamental cellular functions.

## Results

### Proteomics identifies MYCBP2 as a putative EPHB2-interacting protein

To understand the molecular logic underlying EPHB2 signalling diversity, we performed affinity purification coupled to MS (AP-MS) in order to identify EPHB2-interacting proteins, and prioritised those known to be signalling hubs. We used a stable HeLa cell line with tetracycline-inducible expression of BirA-linked EPHB2-FLAG, that we previously used to study EPHB2 signalling (*Lahaie et al., 2019*). To identify ephrin ligand-dependent EPHB2 protein complexes, we stimulated EPHB2-FLAG-overexpressing cells with pre-clustered Fc control or ephrinB2-Fc (eB2-Fc). We then harvested and lysed the cells, performed anti-FLAG immunoprecipitation, and used mass spectrometry to identify EPHB2 protein complexes (*Figure 1A*).

To identify EPHB2-specific interactions and remove background contaminants, we performed Significance Analysis of INTeractome (SAINTexpress analysis *Teo et al., 2014*) using our EPHB2-related controls (*Lahaie et al., 2019*). To better visualize the changes in putative EPHB2 binding partners, we compared the average spectral counts of the identified proteins using ProHits-viz tool (*Knight et al., 2017*). Comparison of Fc and ephrin-B2-treated samples did not yield any significant differences. Since we applied the ligand and collected the samples on a time scale comparable to known EphB signalling dynamics, this limitation could be potentially due to autoactivation of Eph receptors when they are overexpressed, obscuring some ligand-dependent effects (*Lackmann et al., 1998*). However, the resulting scatter plot confirmed the presence of several known EPHB2 interactors, such as FYN and YES1 Src kinases and members of the Eph receptor family (*Figure 1B*; *Banerjee et al., 2022*). One of the most prominent, novel hits was the E3 ubiquitin ligase and signalling hub protein MYCBP2, and its binding partner FBXO45. FBXO45 was previously identified as a putative EPHB2 interacting protein in large-scale, cell-based interactome studies (*Huttlin et al., 2021*; *Salokas et al., 2022*).

### Biochemical validation of MYCBP2 binding to EPHB2

To confirm that MYCBP2 can indeed form a molecular complex with EPHB2, we tested whether endogenous MYCBP2 co-immunoprecipitates (co-IP) with FLAG-tagged EPHB2 in HEK 293T cells. Based on differences in EPHB2 and EPHA3 interactomes, we reasoned that EPHA3 may serve as a negative control for the EPHB2-MYCBP2 association (*Huttlin et al., 2021*). We found that MYCBP2 coprecipitated with affinity-purified EPHB2-FLAG, but not EPHA3 (*Figure 1C*). This confirmed MYCBP2 binding to EPHB2 and suggested MYCPB2 displays EPH receptor subtype specificity. Importantly, the EPHB2-MYCBP2 interaction was reduced by 24.5% and 53% following ephrin-B1 and ephrin-B2 treatment respectively in HeLa EPHB2 cells, suggesting the involvement of MYCBP2 in ephrin-B:EPHB2 signalling (*Figure 1D and E*, eB1-Fc, p=0.1365; eB2-Fc, p=0.0002). To test this interaction in vivo, we performed co-IP using dissociated rat cortical neurons, which further confirmed MCYBP2 association with EPHB2 (*Figure 1F*).

We next asked whether the kinase activity of EPHB2 is required for the formation of the EPHB2-MYCBP2 complex. To test this, we expressed GFP-tagged wild type (WT) EPHB2 or EPHB2 with a kinase-dead mutation (KD) in HeLa cells. We found that MYCBP2 showed comparable coprecipitation with WT and KD EPHB2 (*Figure 1G*). Thus, EPHB2 kinase activity is not required for the formation of MYCBP2-EPHB2 complexes, suggesting that MYCBP2 association with EPHB2 may be a prelude to ephrin-B-evoked EPHB2 signalling.

### FBXO45 enhances the association between MYCBP2 and EPHB2

Our proteomics screen for EPHB2 interactors also identified FBXO45, the F-box protein that forms a ubiquitin ligase complex with MYCBP2 (*Sharma et al., 2014*; *Saiga et al., 2009*). Thus, we initially reasoned that FBXO45 might perform a similar role in the formation of the MYCBP2-EPHB2 complex. We first tested whether FBXO45 binds to EPHB2 by co-expressing MYC-tagged FBXO45 with EPHB2-FLAG or EPHA3-FLAG in HEK293T cells. Co-IP revealed that FBXO45 can associate with EPHB2 but not EPHA3, suggesting that EPHB2, FBXO45, and MYCBP2 could form a ternary complex (*Figure 2A*). To test this idea, we co-expressed EPHB2-FLAG and GFP-MYCBP2 in the presence or absence of MYC-FBXO45 and examined co-IP efficiency between EPHB2 and MYCBP2. Interestingly, FBXO45 enhanced the interaction between EPHB2 and MYCBP2 (*Figure 2B and C*, p=0.0068). Together, these data suggest that EPHB2 can form a complex with both MYCBP2 and FBXO45 and that FBXO45 increases the efficiency of MYCBP2-EPHB2 interaction.

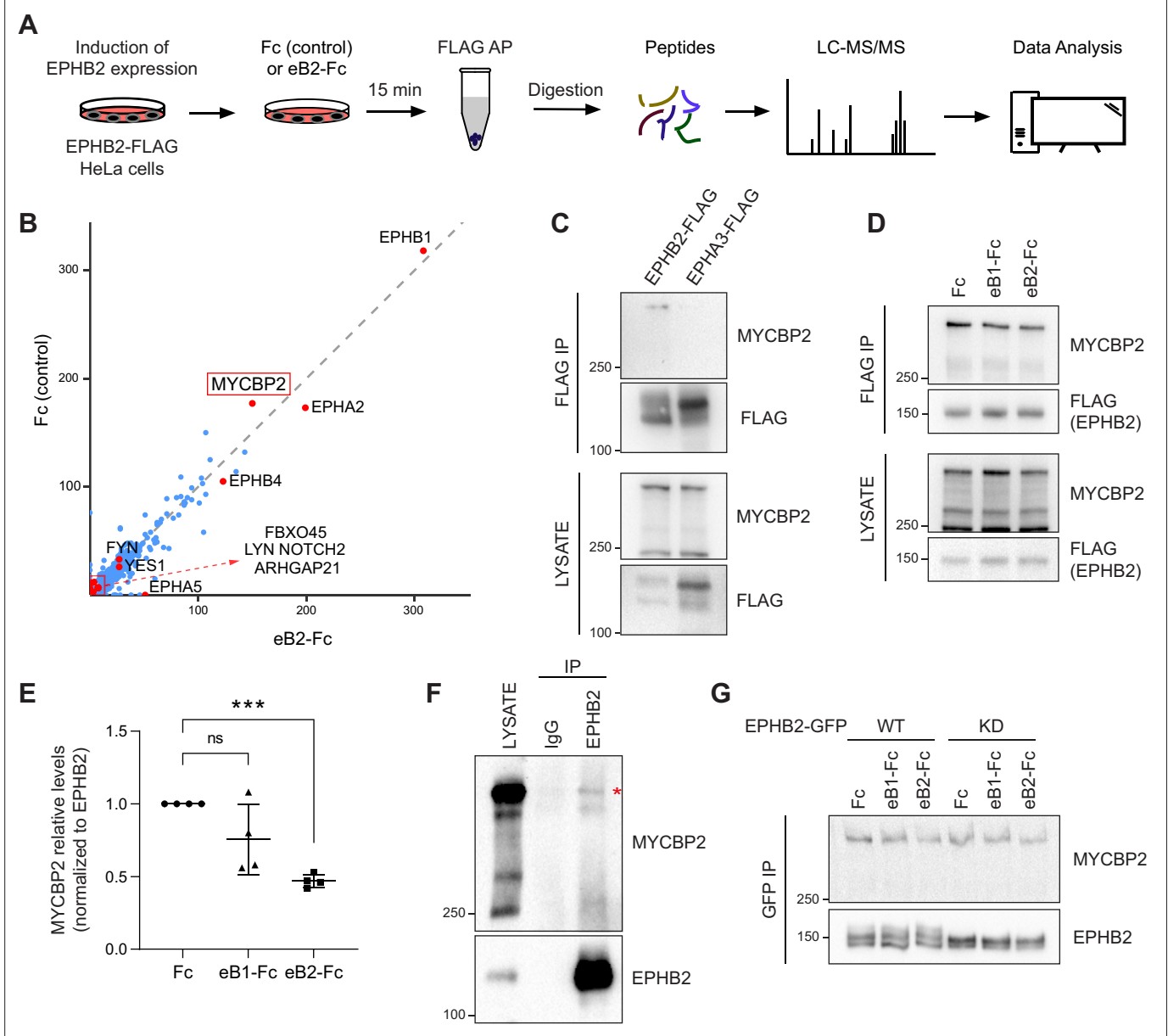

**Figure 1.** MS-proteomics and biochemistry in HeLa cells identifies MYCBP2 as EPHB2 binding protein. (**A**) Schematic of EPHB2 affinity purification coupled to mass spectrometry (AP-MS) workflow. (**B**) Scatter plot of AP-MS data showing known and putative EPHB2 binding proteins, including MYCBP2. Y and X axes represent the average spectral counts of the identified protein hits in the EPHB2 protein complexes from cells stimulated with Fc control or ephrin-B2 (eB2-Fc), respectively. (**C**) In HEK 293T cells, endogenous MYCBP2 is pulled down by transiently overexpressed EPHB2-FLAG but not by EPHA3-FLAG. (**D**) In EPHB2-FLAG stable HeLa cell line, ephrin-B stimulation reduces the interaction between MYCBP2 and EPHB2. (**E**) Quantification of MYCBP2-EPHB2 association intensity after Fc, ephrin-B1 (eB1-Fc) or ephrin-B2 (eB2-Fc) treatment (eB1-Fc, p=0.1365; eB2-Fc, p=0.0002; one-sample t-test). EPHB2-MYCBP2 interaction reduction evoked by eB1-Fc is not statistically significant, probably because of high experimental variability which could be biologically significant. Error bars represent standard deviation (SD). (**F**) Representative image of MYCBP2 pull down with anti-EPHB2 or IgG control antibodies from rat cortical neurons. Asterisk indicates MYCBP2. (**G**) Representative images from western blot analysis of endogenous MYCBP2 following IP of GFP-EPHB2 wild-type (WT) or its kinase dead (KD) counterpart.

The online version of this article includes the following source data for figure 1:

**Source data 1.** Related to *Figure 1C and D*.

**Source data 2.** Related to *Figure 1F and G*.

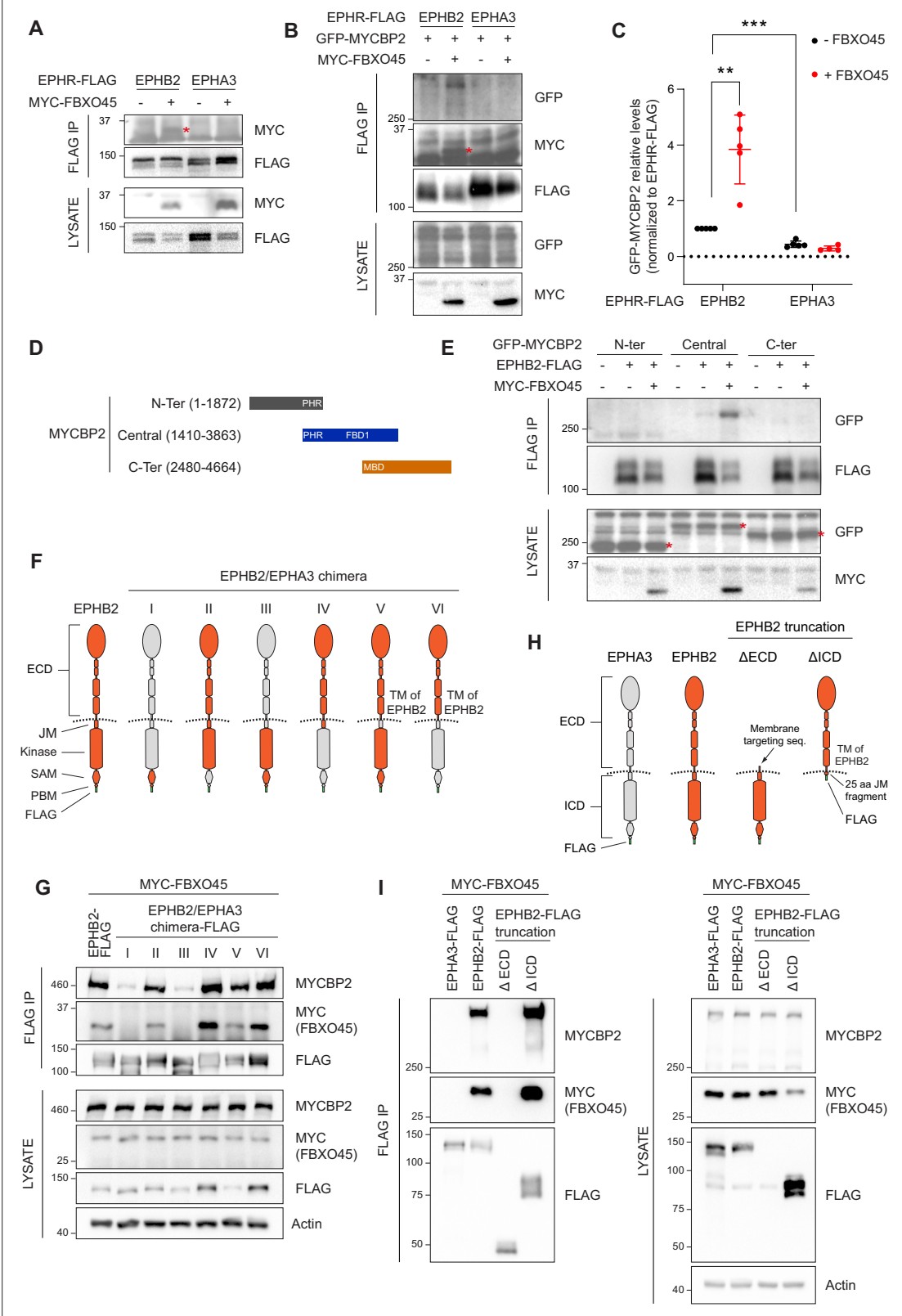

**Figure 2.** Mapping binding regions for EPHB2-MYCBP2 reveals role of FBXO45 in this interaction. (**A**) Co-IP of EPHB2-FLAG with MYC-FBXO45 using transfected HEK293 cells. EPHB2 co-precipitates FBXO45, but EPHA3 does not. Asterisk indicates MYC-FBXO45. (**B**) In HEK 293T cells, FBXO45 overexpression enhances EPHB2-MYCBP2 binding. Asterisk indicates MYC-FBXO45. (**C**) Quantification of the association intensity of MYCBP2 and EPHB2 upon FBXO45 overexpression (EPHB2, p=0.0068; EPHB2 vs EPHA3, p=0.0005; one sample t-test). Error bars represent SD. (**D**) Schematic

*Figure 2 continued on next page*

*Figure 2 continued*

representation of MYCBP2 N-terminal, Central, and C-terminal fragments. (**E**) Co-IP of EPHB2-FLAG with GFP-MYCBP2 fragments in HEK 293T cells. EPHB2 coprecipitates with MYCBP2 central fragment. Asterisks indicate GFP-MYCBP2 fragments. (**F**) Schematic of chimeric domain swapping of EPHB2 (orange) and EPHA3 (grey). (**G**) Co-IP of MYC-FBXO45 and endogenous MYCBP2 with EPHB2/EPHA3 domain swapped chimeras. (**H**) Schematic representation of EPHB2 ΔECD (extracellular domain, aa deletions of 19–530) and ΔICD (intracellular domain, aa deletions of 590–986) truncations. (**I**) Co-IP of endogenous MYCBP2 with EPHA3, EPHB2 and EPHB2 truncation mutants. ECD, extracellular domain; TM, transmembrane; JM, juxtamembrane; SAM, Sterile alpha motif; PBM, PDZ (PSD-95, Dlg1, Zo-1) binding motif.

The online version of this article includes the following source data for figure 2:

**Source data 1.** Related to *Figure 2A, B and E*.

**Source data 2.** Related to *Figure 2G, I*.

## Biochemical mapping of EPHB2-MYCBP2 interaction

To identify the MYCBP2 protein domain(s) required for the formation of the ternary complex with EPHB2 and FBXO45, we co-expressed EPHB2-FLAG and three GFP-MYCBP2 fragments in HEK293T cells. Co-IP revealed that the central region of MYCBP2 was sufficient for binding with EPHB2 (*Figure 2D and E*). In addition, co-expression of MYC-FBXO45 demonstrated that the association of the central domain of MYCBP2 with EPHB2 is enhanced by FBOX45 and is consistent with the presence of an FBXO45 binding site within this MYCBP2 fragment (*Figure 2E*; *Sharma et al., 2014*).

To identify the domains of EPHB2 required for the formation of the tripartite complex, we took advantage of the observation that EPHA3 does not readily form a complex with MYCBP2 or FBXO45. Thus, we performed domain swapping experiments between EPHB2 and EPHA3 reasoning that placing EPHA3-specific sequences in EPHB2 would inhibit the formation of the tripartite complex. We constructed a series of FLAG-EPHB2/EPHA3 chimeras and determined whether they bound MYCBP2 in the presence of FBXO45 (*Figure 2F*). Co-IP revealed that the ability of a particular chimera to associate with MYCBP2 was correlated with its association with FBXO45, in line with the MYCBP2-FBXO45 complex interacting with EPHB2. Surprisingly, we found that EPHB2-EPHA3 chimeras with an EPHA3 identity of intracellular juxtamembrane, kinase, SAM or PDZ binding domains retained the ability to associate with FBXO45 and MYCBP2, suggesting that the formation of the tripartite complex is driven by the extracellular domain and/or the transmembrane domain of EPHB2 (*Figure 2G*). However, these results are also consistent with the possibility that the EPHB2 identity of the extracellular fragments could alter the conformation of the intracellular domains of EPHA3 identity, allowing the interaction with FBXO45-MYCBP2 to occur. To exclude this possibility, we created EPHB2 mutants lacking the intracellular or extracellular domains and tested their ability to complex with MYCBP2 and FBXO45 by co-IP (*Figure 2H*). In these experiments, only the deletion mutant lacking the intracellular domain retained its ability to form the tripartite complex. Collectively, these results argue that the combination of extracellular and transmembrane domains of EPHB2 are necessary and sufficient for formation of the MYCBP2–FBXO45–EPHB2 complex (*Figure 2I*). Since EPHB2 is a transmembrane protein and MYCBP2 is localised in the cytosol, these experiments suggest that the interaction between the extracellular domain of EPHB2 and MYCBP2 might be indirect and mediated by other unknown transmembrane proteins.

## MYCBP2 is required for EPHB2-mediated cellular responses

Given the decrease in MYCBP2–EPHB2 association evoked by ephrin-B treatment (*Figure 1D and E*), we next sought to determine whether MYCBP2 fulfils a specific function in ephrin-B:EPHB2 forward signalling. Thus, we infected EPHB2-FLAG HeLa cells using lentivirus containing CRISPR sgRNA targeting the *MYCBP2* exon 6, and pooled *MYCBP2*^CRISPR cells after puromycin selection (*Figure 3A*). Using EPHB2-FLAG HeLa cells carrying a stably integrated empty expression vector (CTRL^CRISPR) as controls, we found that *MYCBP2*^CRISPR led to a reduction in endogenous MYCBP2 protein levels (*Figure 3B*).

To study the role of MYCBP2 in EphB signalling, we took advantage of an experimental paradigm in which exposure to ephrin-B2 evokes cytoskeletal contraction of HeLa cells expressing EPHB2 (*Lahaie et al., 2019*). Thus, following induction of EPHB2 expression, *MYCBP2*^CRISPR and CTRL^CRISPR cells were stimulated with pre-clustered Fc control or ephrin-B2 for 15 min and scored as collapsed or uncollapsed. While the proportion of collapsed cells for the two cell lines treated

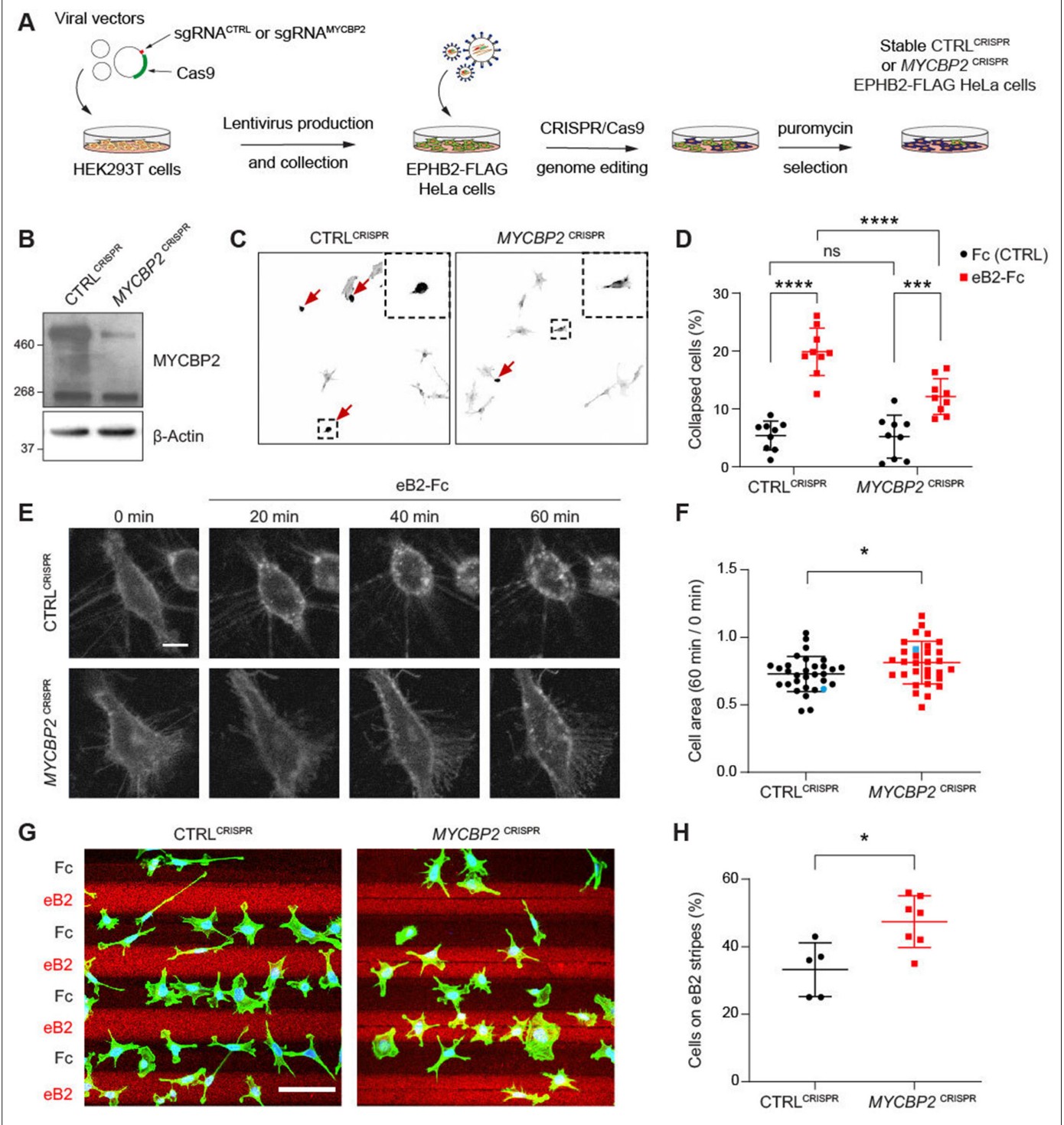

**Figure 3.** MYCBP2 CRISPR HeLa cells exhibit reduced ephrin-B2 evoked cell retraction and ephrin-B2 stripe avoidance. (**A**) Schematic of generation of stable CTRL[CRISPR] or *MYCBP2*[CRISPR] HeLa EPHB2-FLAG cells. Note that these are not clonal cell lines. (**B**) MYCBP2 is reduced in HeLa *MYCBP2*[CRISPR] cells generated by sgRNA targeting MYCBP2 exon 6. (**C**) Representative images of cell collapse assays using CTRL[CRISPR] or *MYCBP2*[CRISPR] HeLa cells that were stimulated with ephrin-B2. Red arrows indicate rounded/collapsed cells. Scale bar is 10 μm, (**D**) Quantification of collapsed cells. Statistical significance between CTRL[CRISPR] and *MYCBP2*[CRISPR] cells was determined using two-way ANOVA followed by Sidak's multiple comparison test (CTRL[CRISPR] vs *MYCBP2* [CRISPR]: Fc, p=0.9903; eB2-Fc, p<0.0001. Fc vs eB2-Fc: CTRL[CRISPR], p<0.0001; *MYCBP2* [CRISPR], p=0.0003). (**E**) Representative time-lapse sequences of CTRL[CRISPR] and *MYCBP2*[CRISPR] HeLa cells after ephrin-B2 treatment. (**F**) Quantification of cell area reduction after 60 min exposure to ephrin-B2. Cell area contraction ratio: CTRL[CRISPR], 27.1%; *MYCBP2*[CRISPR], 18.8%. p=0.0268, two-tailed unpaired t test. Data points corresponding to cells in representative

*Figure 3 continued on next page*

*Figure 3 continued*

images in panel E are in blue. (**G**) Ephrin-B2 stripe assays using CTRL[CRISPR] or *MYCBP2*[CRISPR] HeLa cells. Cells are visualized with Phalloidin 488 staining and nuclei are stained with DAPI (black, Fc stripes; red, ephrin-B2 stripes). Scale bar is 50 μm. (**H**) Quantification of cells present on ephrin-B2 stripes (%). Statistical significance was determined using two-tailed unpaired t-test (p=0.0109). Error bars represent SD.

The online version of this article includes the following source data for figure 3:

**Source data 1.** Related to *Figure 3B*.

with Fc was similar (CTRL[CRISPR], 5.4%; *MYCBP2*[CRISPR], 5.2%; p=0.9903), ephrin-B2 treatment resulted in the collapse of 19.9% of CTRL[CRISPR] cells but only 12.1% of *MYCBP2*[CRISPR] cells (n=9 coverslips; *Figure 3C and D*; p<0.0001). Moreover, when compared to the Fc conditions, *MYCBP2*[CRISPR] cells exhibited a less drastic change in collapse rate upon ephrin-B2 treatment (*Figure 3C and D*; eB2 vs Fc: CTRL[CRISPR], p<0.0001; *MYCBP2*[CRISPR], p=0.0003). In addition, time-lapse imaging of *MYCBP-2*[CRISPR] and CTRL[CRISPR] cells transiently transfected with an EPHB2-GFP expression plasmid revealed a similar attenuation of ephrin-B2-induced cellular contraction (*Figure 3E and F*; p=0.0268). These data argue that MYCBP2 regulates a short-term cellular response evoked by ephrin-B2:EPHB2 signalling.

To study longer-term cellular responses evoked by ephrin-B2:EPHB2 signalling, we turned to a stripe assay in which the preference of CTRL[CRISPR] and *MYCBP2*[CRISPR] cells for immobilized ephrin-B2 or Fc was measured. To do this, cells of either line were deposited over alternating stripes of ephrin-B2 or Fc, and stripe preference was scored for individual cells after overnight incubation. While only 33.2% of CTRL[CRISPR] cells resided on ephrin-B2 stripes, this proportion was significantly increased to 47.4% for *MYCBP2*[CRISPR] cells, suggesting the loss of MYCBP2 function led to a decreased repulsion from ephrin-B2 stripes (*Figure 3G and H*; n=5 and 7 carpets respectively, p=0.0109). When cells were plated on Fc:Fc stripes, CTRL[CRISPR] and *MYCBP2*[CRISPR] cells exhibited no preference over cy3-conjugated Fc stripes (49.18% vs 51.88%, p=0.5386, images not shown). Native HeLa cells only respond to ephrin-B2 once they are made to express EphB2. Thus, our data suggest that MYCBP2 is required for EPHB2-mediated cellular responses in HeLa cells.

## Loss of MYCBP2 decreases cellular levels of EPHB2 protein

The association of EPHB2 with MYCBP2 and its substrate recognition protein FBXO45 suggests that the MYCBP2 ubiquitin ligase complex could target EPHB2 for degradation, a mechanism frequently deployed to terminate transmembrane receptor signalling (*Foot et al., 2017*). However, the results of our cellular assays contradicted this model, and rather suggested that loss of MYCBP2 function decreased EPHB2 signalling. To further evaluate these two scenarios, we first compared EPHB2 protein levels in HeLa *MYCBP2*[CRISPR] cells and CTRL[CRISPR] cells by using tetracycline to induce the EPHB2 overexpression. We found that MYCBP2 loss reduced EPHB2 protein levels (*Figure 4A*). To further confirm this, instead of inducing EPHB2 overexpression with tetracycline, we transfected EPHB2-FLAG plasmid into both CTRL[CRISPR] and *MYCBP2*[CRISPR] cells and examined EPHB2-FLAG levels after two days. As shown in *Figure 4B and C*, levels of EPHB2-FLAG were significantly lowered by 26.1% in HeLa *MYCBP2*[CRISPR] cells compared to CTRL[CRISPR] cells (p=0.0046). We further investigated whether MYCBP2 affects EPHB2 protein turnover when cycloheximide is added to prevent new protein synthesis. EPHB2-FLAG expression was induced by tetracycline for 12 hr followed by protein synthesis inhibition with cycloheximide. We found that EPHB2 half-life was reduced in HeLa cells lacking MYCBP2 compared to control (*Figure 4D and E*; 8 hr treatment, p=0.0474).

Ephrin ligand treatment eventually results in Eph receptor degradation, a process associated with signalling termination. We therefore asked whether ligand-mediated EPHB2 receptor degradation depends on MYCBP2. We induced EPHB2 expression in CTRL[CRISPR] and *MYCBP2*[CRISPR] cells, and exposed them to ephrin-B2 for a different length of time. Western blotting revealed that 4–8 hr stimulation with ephrin-B2 reduced EPHB2 levels in CTRL[CRISPR] cells, but this effect was more drastic in *MYCBP2*[CRISPR] cells (*Figure 4F and G*). Taken together, our results are not consistent with MYCBP2 ubiquitinating EPHB2 and causing its degradation. Unexpectedly, our data indicate that MYCBP2 stabilizes EPHB2 in HeLa cells under both naive and ligand-challenged conditions.

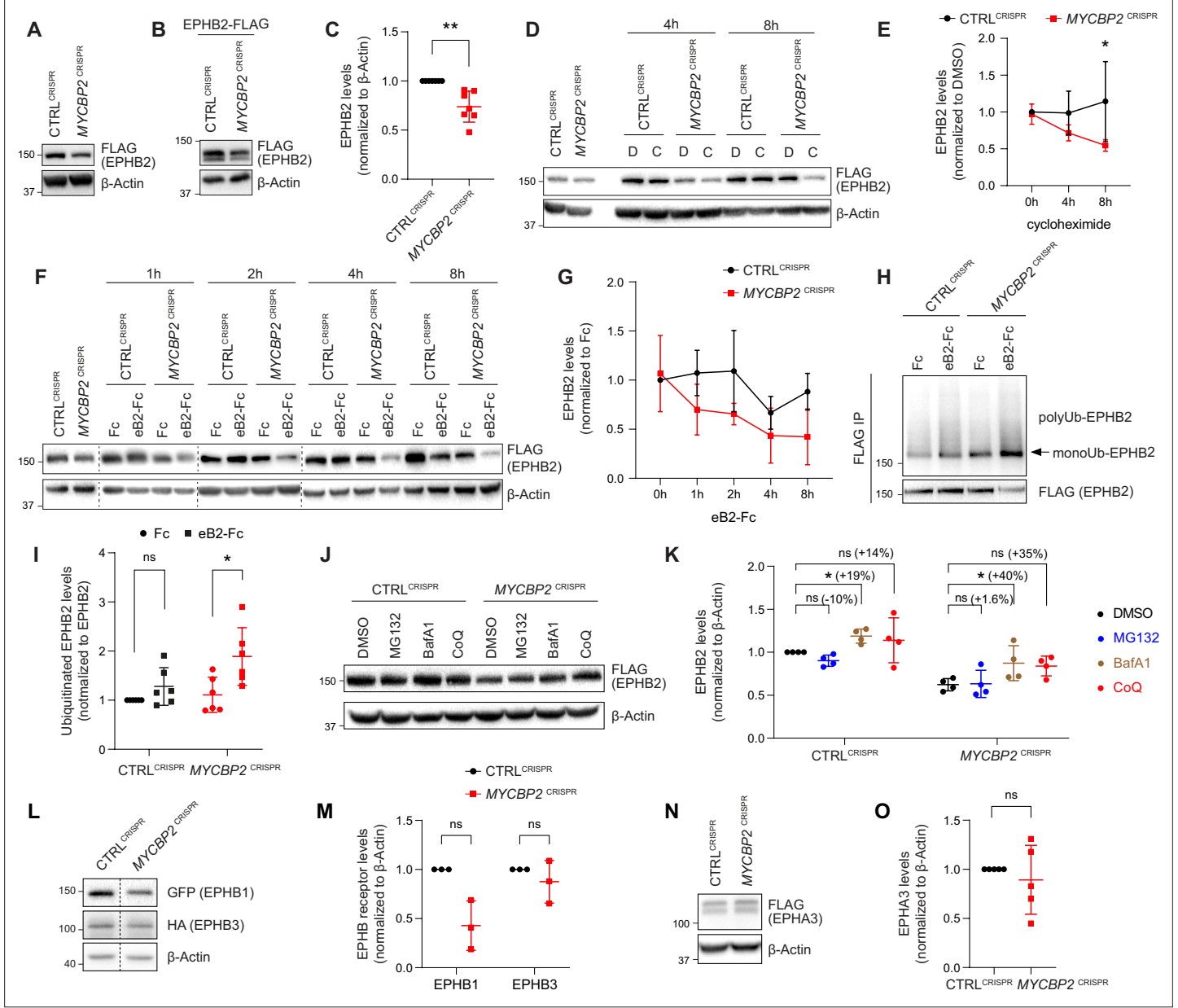

**Figure 4.** MYCBP2 loss-of-function increases EPHB2 protein turnover in HeLa cells. (**A**) Induced EPHB2-FLAG expression is reduced in *MYCBP2*CRISPR HeLa cells. (**B**) Western blotting for transfected EPHB2-FLAG in CTRLCRISPR and *MYCBP2*CRISPR cells. (**C**) Quantification of transfected EPHB2-FLAG levels (p=0.0046, one-sample t-test). (**D**) Representative western blot of EPHB2-FLAG in CTRLCRISPR and *MYCBP2*CRISPR cells treated with DMSO or cycloheximide for 4 hr and 8 hr. (**E**) Quantification of EPHB2-FLAG turnover with cycloheximide (CTRLCRISPR vs. *MYCBP2*CRISPR at 8 hr eB2-Fc stimulation, p=0.0474, two-way ANOVA followed by Tukey's multiple comparison test). (**F**) Western blot showing EPHB2-FLAG degradation when cells are challenged with ephrin-B2 (1 μg/ml) for different periods of time. (**G**) Quantification of ephrin-B2-evoked EPHB2 degradation (ns, not significant; two-way ANOVA followed by Tukey's multiple comparison test). Although not significant, there is an apparent trend towards lower EPHB2 levels in *MYCBP2*CRISPR cells, which could become significant with additional replicates. (**H**) Western blot of EPHB2 ubiquitination in CTRLCRISPR and *MYCBP2*CRISPR cells. (**I**) Quantification of ubiquitinated EPHB2. CTRLCRISPR cells stimulated with Fc vs. eB2-Fc, p=0.1349 (One sample t-test); *MYCBP2*CRISPR cells stimulated with Fc vs. EB2-Fc, p=0.0195 (Unpaired two-tailed t-test). (**J**) After tetracycline induction of EPHB2-FLAG expression for 16 hr, CTRLCRISPR and *MYCBP2*CRISPR HeLa cells were treated with DMSO (1:500) or inhibitors of the proteasome (MG132 50 μM) or lysosome (BafA1 0.2 μM; CoQ 50 μM) for 6 hr, and EPHB2 levels were analysed by western blotting. (**K**) Quantification of EPHB2 levels following treatment with proteasome or lysosome inhibitors. Statistical significance for the comparison between CTRLCRISPR cells treated with DMSO or inhibitors was determined by one-sample t-test (MG132, p=0.0598; BafA1, p=0.0200; CoQ, p=0.3632), whereas statistical significance for the comparison between *MYCBP2*CRISPR cells treated with DMSO and individual inhibitors was determined by two-tailed paired t-test (MG132, p=0.8893; BafA1, p=0.0361; CoQ, p=0.0835). (**L**) GFP-EPHB1 and HA-EPHB3 transfected into CTRLCRISPR and *MYCBP2*CRISPR HeLa cells and detected by western blot. (**M**) Quantification of GFP-EPHB1 and HA-EPHB3 levels (EPHB1, p=0.0588;

*Figure 4 continued on next page*

*Figure 4 continued*

EPHB3, p=0.4253; one-sample t-test). (**N**) FLAG-EPHA3 transfected into CTRL[CRISPR] and *MYCBP2*[CRISPR] HeLa cells and detected by WB. (**O**) Quantification of FLAG-EPHA3 (p=0.5369, one-sample t-test). Error bars represent SD.

The online version of this article includes the following source data for figure 4:

**Source data 1.** Related to *Figure 4A, B and D*.

**Source data 2.** Related to *Figure 4F*.

**Source data 3.** Related to *Figure 4H and J*.

**Source data 4.** Related to *Figure 4L*.

## Loss of MYCBP2 enhances ligand-induced EPHB2 receptor ubiquitination

Ubiquitination of receptor tyrosine kinases, including Eph receptors, can herald their degradation via the proteasome and thus termination of signalling (*Haglund and Dikic, 2012*; *Sabet et al., 2015*). This model is not supported by our results, which suggest that MYCBP2 is required for EPHB2 protein maintenance. Nonetheless, we investigated whether EPHB2 receptor ubiquitination is altered in HeLa cells depleted of MYCBP2. HeLa CTRL[CRISPR] and *MYCBP2*[CRISPR] cells were transfected with HA-tagged ubiquitin, EPHB2 expression was induced with tetracycline, and cells were treated with ephrin-B2 for 30 min. We observed that EPHB2 receptor ubiquitination was not significantly increased in CTRL[CRISPR] cells after this short-term ligand treatment (*Figure 4H, I* CTRL[CRISPR], *P*=0.1349). In contrast, EPHB2 ubiquitination was significantly increased in *MYCBP2*[CRISPR] cells (*Figure 4H, I* MYCBP2[CRISPR], *P*=0.0195). This effect argues against the concept that EPHB2 is a MYCBP2 ubiquitination substrate, and suggests that in the absence of MYCBP2 degradation of the EPHB2 receptor is enhanced due to increased ubiquitination.

## A potential involvement of the lysosomal pathway in EPHB2 degradation caused by the loss of MYCBP2

EPHB2 can be degraded by either a proteasomal or lysosomal pathway depending on the cellular context (*Cissé et al., 2011*; *Litterst et al., 2007*; *Fasen et al., 2008*). Thus, we wanted to shed light on how EPHB2 is degraded and understand why EPHB2 degradation is enhanced by MYCBP2 loss of function. To do so, we induced EPHB2 expression in CTRL[CRISPR] and *MYCBP2*[CRISPR] cells and applied the S26 proteasome inhibitor MG132, or the lysosomal inhibitors BafilomycinA1 or Chloroquine. We found that MG132 did not have significant effects on EPHB2 levels in both cell types (*Figure 4J and K*). However, we found that BafilomycinaA1 (BafA1) significantly increased EPHB2 protein levels in both HeLa CTRL[CRISPR] cells and *MYCBP2*[CRISPR] cells by 19% and 40%, respectively (*Figure 4J and K*). We also observed a trend towards increased EPHB2 levels with Chloroquine (CoQ) treatment in CTRL[CRISPR] (14%) and *MYCBP2*[CRISPR] (35%), further suggesting a role for lysosomal degradation (*Figure 4J and K*). Although the difference in percentage increase between CTRL[CRISPR] cells and *MYCBP2*[CRISPR] cells is not significant, this trend raises the possibility that the loss of MYCBP2 promotes EPHB2 receptor degradation through the lysosomal pathway.

## Regulation of Eph receptor levels by MYCBP2

The above experiments raise the question of whether MYCBP2 is a general regulator of Eph receptor stability. Since EPHA3 does not form a complex with MYCBP2, and EphA receptor levels are controlled by the proteasomal pathway (*Sharfe et al., 2003*; *Walker-Daniels et al., 2002*), we hypothesized that MYCBP2 might regulate the levels of the entire EphB receptor class. To test this, we co-transfected plasmids encoding GFP-tagged EPHB1 and HA-tagged EPHB3 into HeLa CTRL[CRISPR] and *MYCBP2*[CRISPR] cells. Compared to CTRL[CRISPR] cells, EPHB1 and EPHB3 levels were reduced by 57.1% and 12.5% respectively in *MYCBP2*[CRISPR] cells (*Figure 4L and M*, EPHB1, p=0.0588; EPHB3, p=0.4253). Although not statistically significant, there is an apparent trend towards a decrease in EPHB1 levels. On the other hand, FLAG-EPHA3 levels were similar in CTRL[CRISPR] and *MYCBP2*[CRISPR] cells (*Figure 4N and O*, p=0.5369). Taken together, these data suggest that MYCBP2 may stabilize other EphB subclass receptors.

## Loss of MYCBP2 attenuates the magnitude of EPHB2 cellular signalling

EPHB2 receptor activation evokes signal transduction events such as tyrosine phosphorylation of EPHB2, activation of EPHB2 tyrosine kinase function and phosphorylation of the ERK1/2 downstream effector (*Poliakov et al., 2008*). We therefore measured EPHB2 and ERK1/2 phosphorylation in CTRL$^{CRISPR}$ and *MYCBP2*$^{CRISPR}$ cells for up to 8 hours after ephrin-B2 application (*Figure 5A*). P-EPHB2 (pY20) and p-ERK1/2 signals were normalised to GAPDH and ERK1/2, respectively. In HeLa CTRL-$^{CRISPR}$ cells, pTyr-EPHB2 response reached a plateau after 1–2 h treatment and remained up 8 hours post-stimulation (*Figure 5A–C*). On the contrary, ephrin-B2-evoked phosphorylation of EPHB2 was reduced in *MYCBP2*$^{CRISPR}$ cells with quantitative results showing a significant reduction by 8 hours of treatment (*Figure 5A and C*; 8 h *P*=0.0331). We were also able to detect significantly lower p-ERK1/2 levels in *MYCBP2*$^{CRISPR}$ cells relative to CTRL$^{CRISPR}$ cells, although activation of ERK1/2 by ephrin-B2 was variable (*Figure 5D*. 4 h, *P*=0.0494; 8 h, *P*=0.0078; n=6). We again noted significantly enhanced EPHB2 degradation in *MYCBP2*$^{CRISPR}$ cells (*Figure 5E*; 8 h, *P*=0.0437). This decrease was also observed when CTRL$^{CRISPR}$ and *MYCBP2*$^{CRISPR}$ cells were treated with ephrin-B1 (data not shown).

Next, we asked whether the decrease in ligand-evoked EPHB1/2 and ERK1/2 activation in *MYCBP-2*$^{CRISPR}$ cells reflects a requirement for MYCBP2 in the EPHB2 signalling cascade per se, or whether it is explained by decreased EPHB2 protein levels caused by MYCBP2 loss. We thus normalised p-EPHB2 and p-ERK1/2 signal to the levels of EPHB2 protein at all time points, which revealed that kinetics and magnitude of EPHB2 and ERK1/2 phosphorylation are similar between CTRL$^{CRISPR}$ and *MYCBP2*$^{CRISPR}$ cells (*Figure 5F and G*). Although these data argue against a direct role of MYCBP2 in the early events of EPHB2 signalling, they nevertheless indicate that MYCBP2 loss results in marked attenuation of EPHB2 activation and its downstream pERK1/2 signalling in line with decreased cellular responses to ephrin-B2.

## Exogenous Fbxo45 binding domain of MYCBP2 disrupts the EPHB2-MYCBP2 interaction

Previous studies showed that FBXO45 binds to the FBD1 domain of MYCBP2, and exogenous FBD1 overexpression can disrupt the FBXO45-MYCBP2 association (*Sharma et al., 2014*). We thus tested whether FBD1 overexpression can interfere with the formation of the EPHB2-MYCBP2 complex (*Figure 6A*). Indeed, the expression of GFP-FBD1 wild-type (WT) in HEK cells expressing EPHB2-FLAG and MYC-FBXO45 reduced binding between EPHB2 and MYCBP2 (*Figure 6B*). This effect was not observed with a GFP-FBD1 mutant (mut) fragment that harbours three point mutations that inhibit binding to FBXO45. Reduction of the EPHB2-MYCBP2 interaction was also observed in cells that were not overexpressing FBXO45 (*Figure 6C*). Thus, exogenous FBD1 can specifically disrupt the EPHB2-MYCBP2 association. This suggests that FBXO45 binding to MYCBP2 may facilitate for formation of the MYCBP2-EPHB2 complex.

## FBD1 expression impairs EPHB2-mediated neuronal responses

To determine whether the EPHB2-MYCBP2 interaction is required for EPHB2 function, we disrupted binding via FBD1 ectopic expression in cell lines or primary neurons, and studied cellular responses to ephrin-B treatment. We introduced GFP-FBD1 mut or GFP-FBD1 WT into HeLa cells with inducible EPHB2 expression and cultured these cells on ephrin-B2 and Fc stripes. Following overnight culture, only 19.2% of cells expressing GFP-FBD1 mut resided on ephrin-B2 stripes. In contrast, 28.6% of cells expressing GFP-FBD1 WT were found on ephrin-B2 stripes indicating that FBD1 expression can dampen ephrin-B2:EPHB2 mediated cell repulsion (*Figure 6D and E*; *P*=0.0107). In contrast, cells expressing GFP-FBD1 mut or GFP-FBD1 WT displayed no preference for either one of Fc:Fc control stripes (49.63% vs 49.85%, *P*=0.9560, images not shown).

Since EPHB2 is expressed in the embryonic chicken spinal cord, we performed in ovo electroporation to introduce plasmids encoding GFP-FBD1 mut or GFP-FBD1 WT into the spinal cords of 2.5 day old embryonic chickens (Hamburger-Hamilton; (HH) Stage 15–17; *Hamburger and Hamilton, 1951*; *Luria et al., 2008*). Two days later spinal cords were dissected, divided into explants, cultured overnight on alternating stripes containing ephrin-B2 or Fc, and axonal GFP signal present over ephrin-B2 versus Fc stripes was determined. When explants expressed GFP-FBD1 mut (negative control), we observed that 26.5% of neurite GFP signal resided on ephrin-B2 stripes (*Figure 7A and B*). In contrast, we found a significant increase of 32.2% of neurite GFP signal expressing FBD1 WT on ephrin-B2

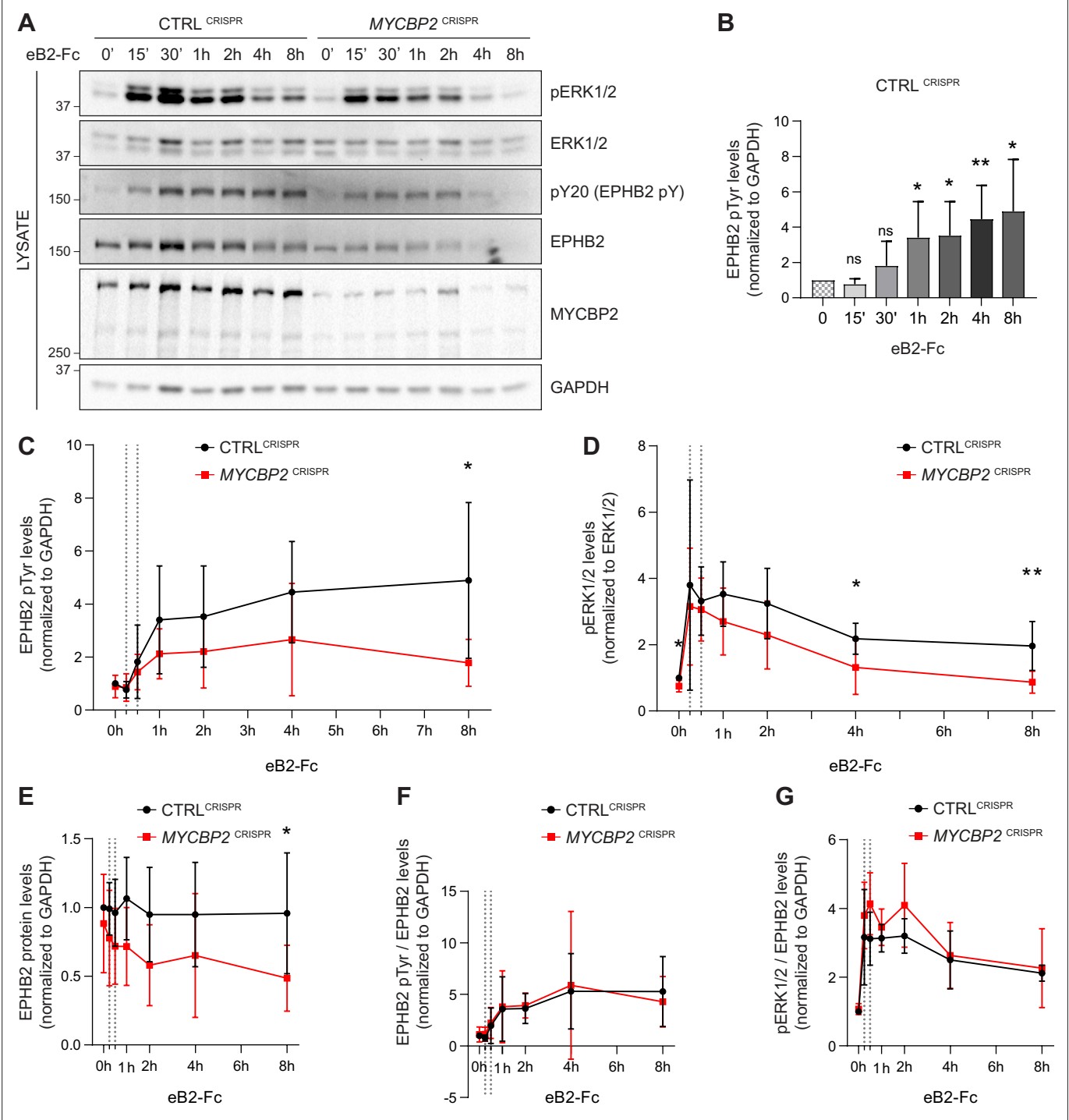

**Figure 5.** MYCBP2 depletion impairs EPHB2 phosphorylation and ERK1/2 activation in HeLa cells. (**A**) Representative western blot for pERK1/2 and pTyr-EPHB2 detected in CTRL[CRISPR] and *MYCBP2*[CRISPR] cells treated with ephrin-B2 (eB2-Fc) for different periods (n=6). Membranes were striped and reblotted with anti-ERK1/2, anti-EPHB2, anti-GAPDH and anti-MYCBP2 antibodies as controls. (**B**) Quantification of EPHB2 tyrosine phosphorylation in CTRL[CRISPR] cells evoked by ephrin-B2 treatment (15 min, $P=0.1363$; 30 min, $P=0.2056$; 1 h, $P=0.0342$; 2 h, $P=0.0234$; 4 h, $P=0.0068$; 8 h, $P=0.0231$; one-sample t-test). (**C**) Quantification of EPHB2 tyrosine phosphorylation in CTRL[CRISPR] and *MYCBP2*[CRISPR] HeLa cells (unstimulated, $P=0.5589$, one-sample t-test; stimulated for 15 min, $P=0.7463$; 30 min, $P=0.5520$; 1 h, $P=0.1920$; 2 h, $P=0.2009$; 4 h, $P=0.1550$; 8 h, $P=0.0331$; two-tailed unpaired t-test). (**D**) Quantification of pERK1/2 in CTRL[CRISPR] and *MYCBP2*[CRISPR] HeLa cells (0 min, $P=0.0168$, one-sample t-test; 15 min, $P=0.6695$; 30 min, $P=0.6649$; 1 h, $P=0.1776$; 2 h, $P=0.1479$; 4 h, $P=0.0494$; 8 h, $P=0.0078$; two-tailed unpaired t-test). (**E**) Quantification of EPHB2 in CTRL[CRISPR] and *MYCBP2*[CRISPR] HeLa (0 min, $P=0.4604$, one-sample t-test; 15 min, $P=0.2222$; 30 min, $P=0.1376$; 1 h, $P=0.0651$; 2 h, $P=0.0736$; 4 h, $P=0.2451$; 8 h, $P=0.0437$, two-tailed unpaired

*Figure 5 continued*

t-test). (**F**) Quantification of ephrin-B2-evoked EPHB2 tyrosine phosphorylation levels relative to total EPHB2 protein levels (0 min, $P=0.7058$, one-sample t-test; 15 min, $P=0.2464$; 30 min, $P=0.7835$; 1 h, $P=0.9164$; 2 h, $P=0.7196$; 4 h, $P=0.8625$; 8 h, $P=0.5750$, two-tailed unpaired t-test). (**G**) Quantification of ephrin-B2-evoked pERK1/2 relative to EPHB2 total protein levels 0 min, $P=0.3308$, one-sample t-test; 15 min, $P=0.3856$; 30 min, $P=0.0624$; 1 h, $P=0.2683$; 2 h, $P=0.1284$; 4 h, $P=0.7998$; 8 h, $P=0.7790$, two-tailed unpaired t-test. Error bars represent SD.

The online version of this article includes the following source data for figure 5:

**Source data 1.** Related to *Figure 5A*.

stripes (*Figure 7A and B*; p=0.0410). Thus, FBD1 expression impairs long-term repulsive responses to ephrinB2-EPHB2 signalling in spinal explants.

To study short-term neuronal responses to ephrin-Bs, we turned to mouse hippocampal neurons and a growth cone collapse assay (*Srivastava et al., 2013*). Here, we electroporated GFP-FBD1 mut or GFP-FBD1 WT expression plasmids into dissociated hippocampal neurons and treated them with pre-clustered Fc or ephrin-B1 for one hour (*Figure 7C*). Neurons expressing GFP-FBD1 mut showed 22.7% growth cone collapse with Fc treatment, while ephrin-B1 treatment elicited significant increases to 35.9% collapse (*Figure 7C and D*, p=0.0006). In contrast, ephrin-B1 failed to significantly induce growth cone collapse in neurons expressing GFP-FBD1 WT: with only 31% growth cones being collapsed by ephrin-B1, compared to 25.3% being collapsed by Fc treatment (*Figure 7D*; p=0.1341). Together, these data indicate that impairing MYCBP2 function via FBD1 expression disrupts ephrin:B-EPHB2 signalling in axonal guidance.

## Genetic interactions between the *C. elegans* Eph receptor and the MYCBP2 signalling network

Next, we sought to test genetic interactions between an ephrin receptor and the MYCBP2 signalling network using an in vivo animal model. To do so, we turned to *C. elegans* which has a sole Eph family receptor (EPHR) called VAB-1 and a single MYCBP2 ortholog called RPM-1 (*Grill et al., 2016*; *George et al., 1998*). Previous studies have shown that RPM-1/MYCBP2 is required to terminate axon outgrowth (*Borgen et al., 2017*; *Schaefer et al., 2000*). Furthermore, RPM-1 functions as a hub upstream of a number of signalling pathways (*Grill et al., 2007*; *Grill et al., 2012*; *Tulgren et al., 2014*; *Baker et al., 2014*). Genetic results from these studies demonstrated that mutants for RPM-1 binding proteins display genetic enhancer interactions with one another, but do not enhance defects when combined with *rpm-1* mutants.

Given the biochemical interactions between MYCBP2, FBXO45 and EPHB2, we first evaluated genetic interactions between VAB-1/EPHR and two known RPM-1 binding proteins: (1) Rab GEF GLO-4, an orthologue of mammalian SERGEF that functions via the GLO-1/RAB32 small GTPase and is not involved in RPM-1 ubiquitin ligase activity (*Grill et al., 2007*). (2) FSN-1, an orthologue of FBXO45, that is the F-box substrate selector protein of the RPM-1 ubiquitin ligase complex (*Figure 8A*; *Liao et al., 2004*). RPM-1 genetic interactions were studied in the left and right PLM mechanosensory neurons of *C. elegans,* both of which extend an axon anteriorly until it terminates posterior to the cell body of the respective ALM mechanosensory neuron (*Figure 8B*). This process is visualized using a transgene, *muIs32 (Pmec-7::GFP)*, that expresses GFP in the PLM and ALM mechanosensory neurons. As observed previously, a null allele of *vab-1* that deletes exons 1–4 showed a significant increase in incidence of PLM axon extension beyond the ALM cell body (overextension) compared to wild-type controls (*Figure 8B and C*; *Mohamed and Chin-Sang, 2006*). To study the interaction between VAB-1/EPHR and GLO-4/SERGEF, we first evaluated *glo-4* mutants, in which we observed two kinds of overextension defects: one where PLM axons extend past the ALM cell body in a straight line, and a more severe defect where PLM axons 'hook' ventrally (*Figure 8B and C*). The frequency of both types of axon termination defects were significantly enhanced in *vab-1; glo-4* double mutants (*Figure 8B and C*). Overextension defects were significantly rescued by transgenic expression of VAB-1 in *vab-1; glo-4* double mutants (*Figure 8C*). Similarly, we observed an enhanced frequency of both overextension and hook defects in *vab-1; fsn-1* double mutants compared to either mutant alone (*Figure 8D*). Thus, VAB-1/EPHR interacts genetically with two proteins known to bind and function downstream of RPM-1/MYCBP2.

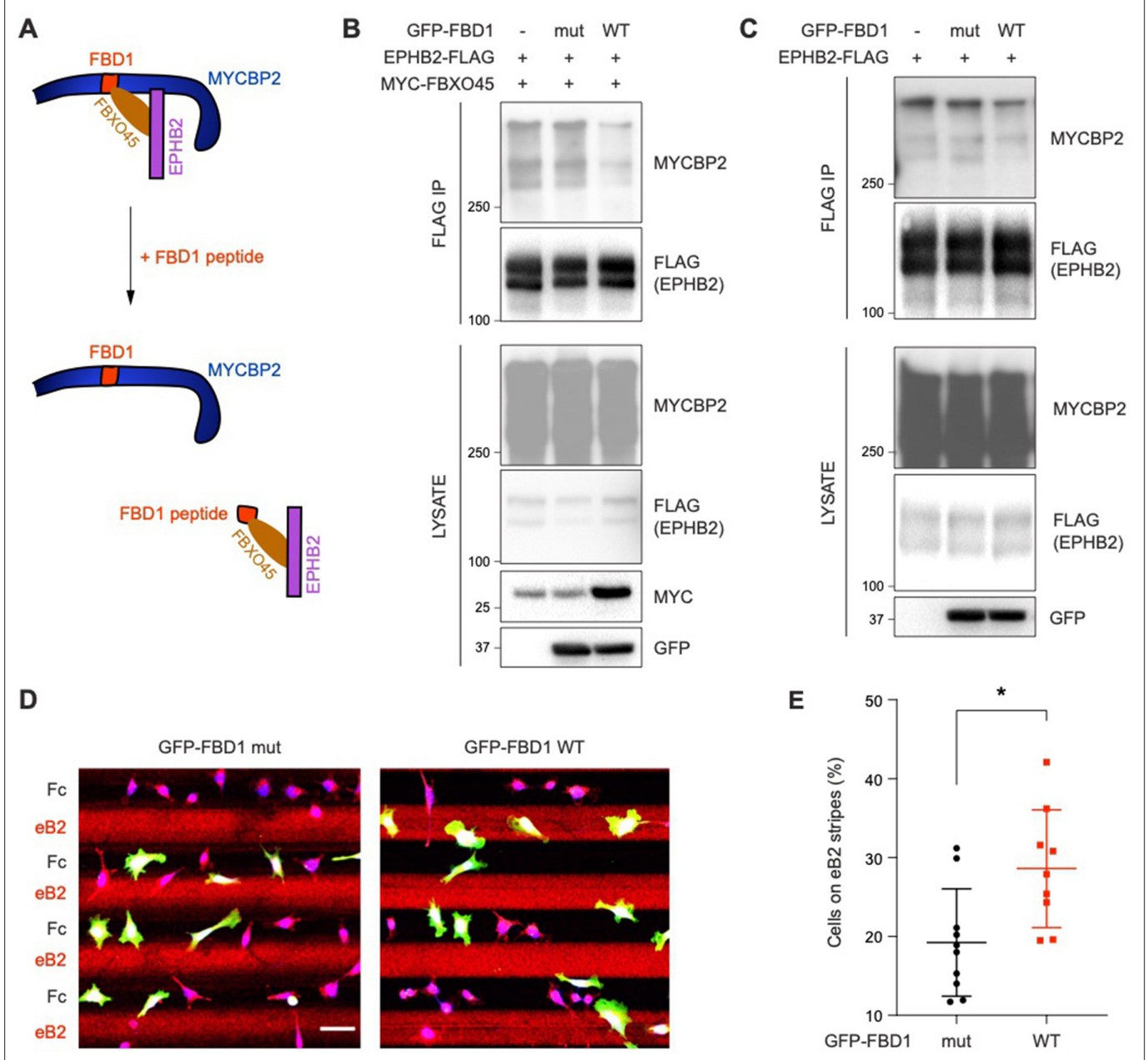

**Figure 6.** Exogenous FBD1 fragment of MYCBP2 disrupts EPHB2-MYCBP2 binding and impairs EPHB2 function in HeLa cells. (**A**) Schematic illustrating competition of exogenous MYCBP2-FBD1 fragment that disrupts MYCBP2-FBXO45 binding and leads to MYCBP2 reduction in EPHB2 complexes. (**B**) Exogenous FBD1 WT overexpression leads to reduced EPHB2-MYCBP2 binding in HEK293 cells despite co-expression of FBXO45. (**C**) FBD1 overexpression also disrupts EPHB2-MYCBP2 binding in the absence of FBXO45 overexpression. (**D**) Representative images of ephrin-B2 stripe assays using HeLa cells expressing GFP-FBD1 mut or GFP-FBD1 WT. Scale bar is 50 μm. (**E**) Quantification of cells present on eB2 stripes (*P*=0.0107, two-tailed unpaired t-test). Error bars represent SD.

The online version of this article includes the following source data for figure 6:

**Source data 1.** Related to *Figure 6B*.

**Source data 2.** Related to *Figure 6C*.

Next, we examined genetic interactions between *vab-1* and an *rpm-1* protein null allele (*AlAbdi et al., 2023*). We compared the incidence of PLM overextension in *vab-1; rpm-1* double mutants to *rpm-1* single mutants and did not observe a significant difference between them (*Figure 8E*). Because *rpm-1* mutants displayed a high frequency of PLM hook defects when the PLM axon was

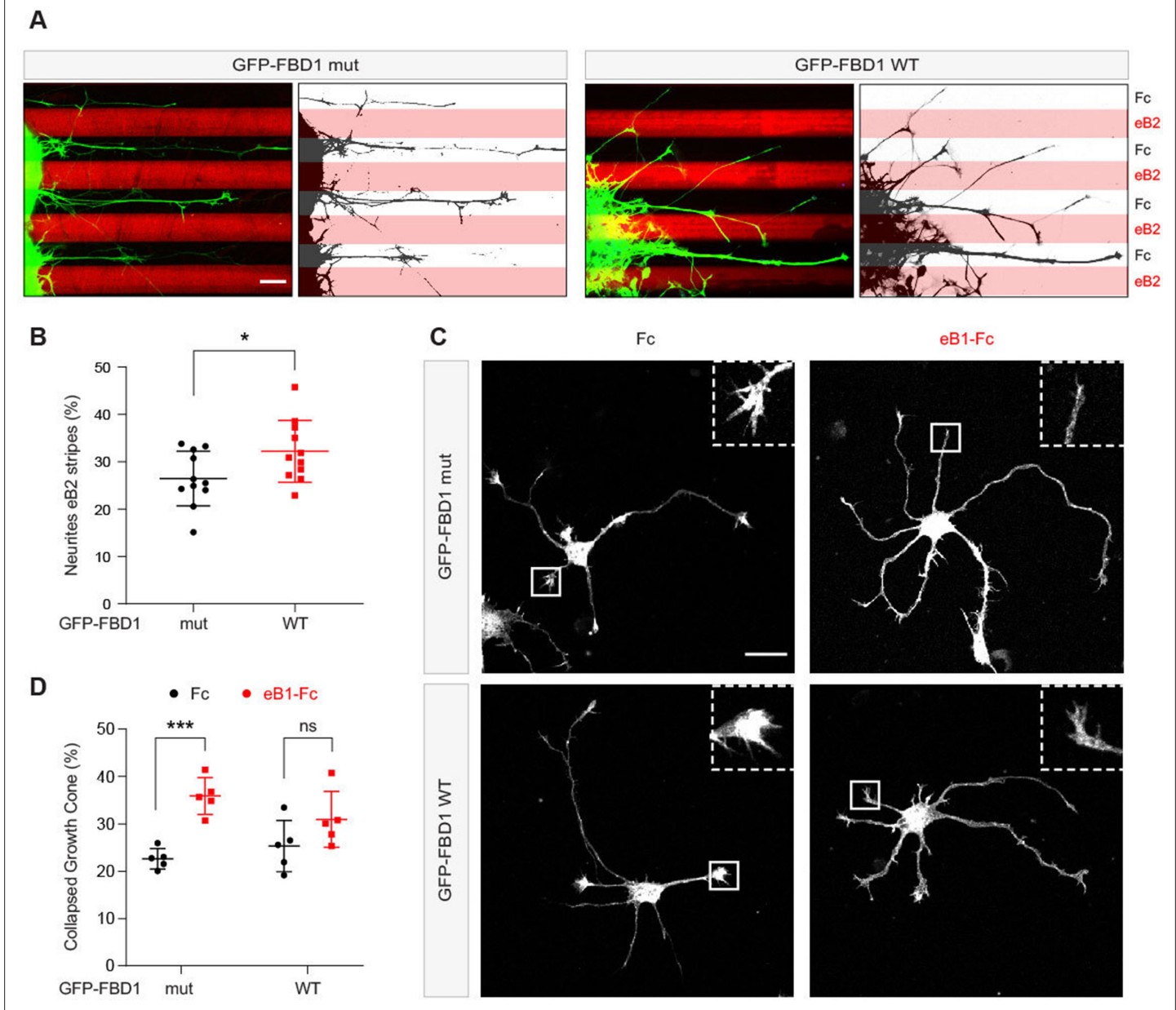

**Figure 7.** Exogenous FBD1 overexpression impairs EPH receptor functions in chick spinal cord explants and mouse hippocampal neurons. (**A**) Representative images of ephrin-B2 stripe assays with chick embryonic spinal cord explants overexpressing GFP-FBD1 mut (negative control) or GFP-FBD1 WT. Images with inverted GFP signal in dark pixels on Fc / eB2 (pink) stripes are placed beside the original images. Scale bar is 50 μm. (**B**) Quantification of GFP-positive neurites present on ephrin-B2 stripes (GFP-FBD1 mut vs. GFP-FBD1 WT, p=0.0410, two-tailed unpaired t-test). (**C**) Representative images of DIV2 mouse hippocampal neurons overexpressing GFP-FBD1 mut or GFP-FBD1 WT and challenged with Fc control or ephrin-B1 (eB1-Fc). Scale bar is 20 μm. (**D**) Quantification of growth cone collapse rate for hippocampal neurites. GFP-FBD1 mut: Fc vs. eB1-Fc, p=0.0006; GFP-FBD1 WT: Fc vs. eB1-Fc p0.1341. Two-way ANOVA followed by Sidak's multiple comparisons test. Error bars represent SD.

visualized using the *muIs32* transgene, we pivoted to address phenotypic saturation. To do so, we evaluated *vab-1* genetic interactions using the *zdIs5 (Pmec4::GFP)* transgene to label PLM axons. This axonal reporter was used previously to demonstrate that the frequency of *rpm-1* hook defects can be enhanced by mutations in genes that are not RPM-1/MYCBP2-binding proteins (*Borgen et al., 2017*). In contrast, the incidence of *rpm-1* hook defects was not increased by mutations that impair RPM-1 binding proteins. Like prior findings, *rpm-1* mutants on the *zdIs5* background result in a lower frequency of hook defects than *rpm-1* mutants on *muIs32* (*Figure 8F*). Using the *zdIs5* background, we found that *vab-1; fsn-1* double mutants display a higher frequency of overextension defects when

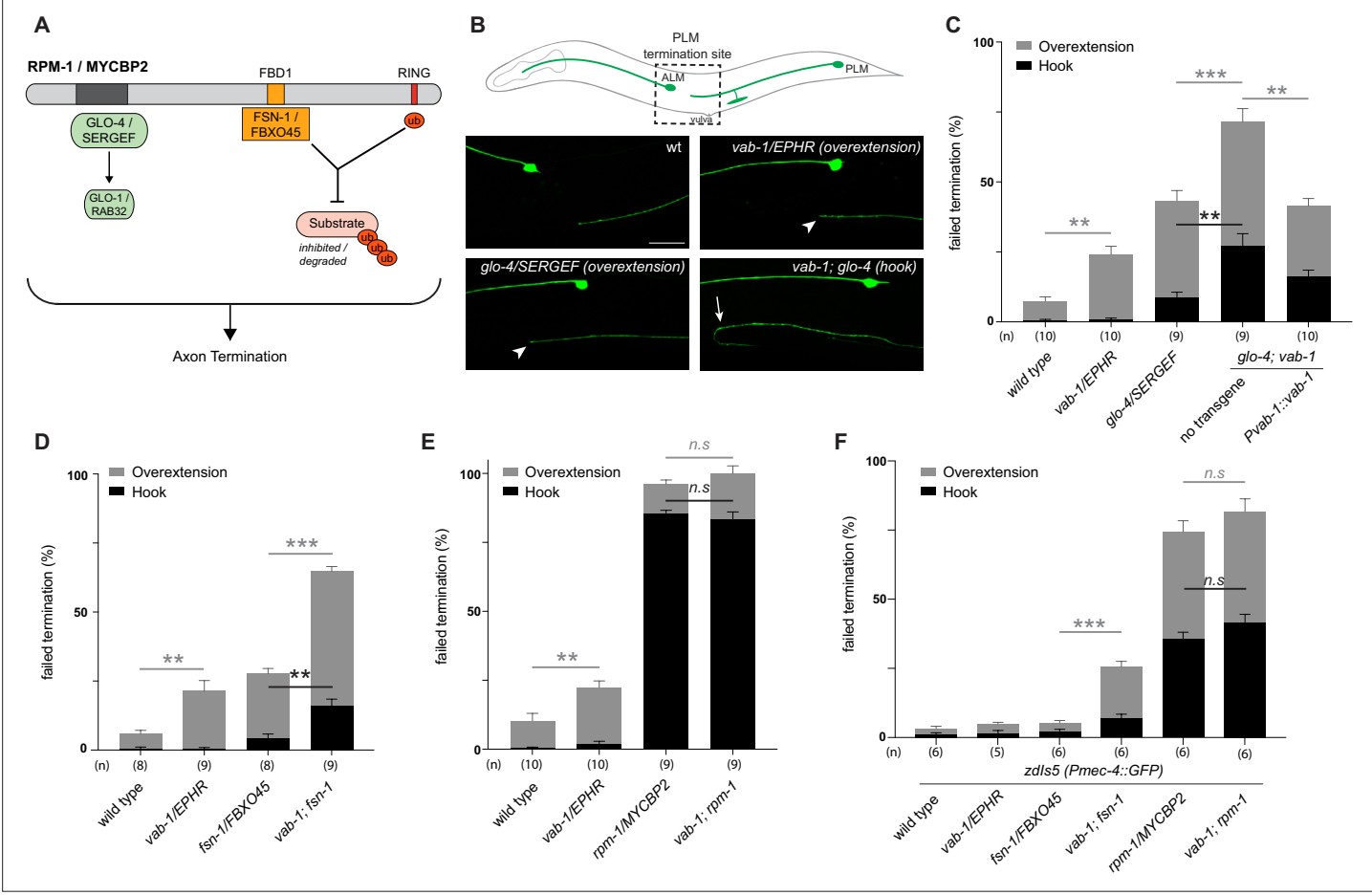

**Figure 8.** *C. elegans VAB-1* ephrin receptor interacts genetically with known *RPM-1/MYCBP2* binding proteins *FSN-1/FBXO45* and *GLO-4/SERGEF*. (**A**) schematic showing the known RPM-1/MYCBP2 binding proteins GLO-4/SERGEF and FSN-1/FBXO45. GLO-4 functions independent of RPM-1 ubiquitin ligase. FSN-1 is the F-box protein that forms a ubiquitin ligase complex with RPM-1. Adapted from ***Grill et al., 2016***. (**B**) Schematic representation of axon morphology and axon termination site for PLM mechanosensory neurons and representative images of failed axon termination defects observed in PLM neurons for indicated genotypes. Axon termination visualized using *muIs32 (Pmec-7::GFP)*, which expresses GFP in the PLM and ALM mechanosensory neurons. Examples of moderate severity overextension defects (arrowhead) observed in *vab-1/EphR* and *glo-4/SERGEF* single mutants. Example of severe overextension (hook) defects (arrow) observed in *vab-1; glo-4* double mutants. (**C**) Quantitation of axon termination defects for indicated genotypes using *muIs32. vab-1; glo-4* double mutants show enhanced frequency of both hook (black) and overextension (grey) failed termination defects. Overextension defects are significantly reduced by transgenic expression of VAB-1. (**D**) Quantitation of axon termination defects for indicated genotypes. *vab-1; fsn-1* double mutants show enhanced termination defects. (**E**) Quantitation of axon termination defects for indicated genotypes using *muIs32*. Axon termination defects are not suppressed in *vab-1; rpm-1* double mutants compared to *rpm-1* single mutants. (**F**) *zdIs5 (Pmec-4::GFP)* was used to quantify axon termination defects for indicated genotypes. *vab-1; fsn-1* double mutants show enhanced frequency of overextension defects (grey). Frequency and severity of axon termination defects is not significantly different between *vab-1; rpm-1* double mutants and *rpm-1* single mutants. n is defined as a single count of 20–30 animals. Means are shown from 8 to 10 counts (20–30 animals per count) for each genotype, and error bars represent SEM. Significance determined using Student's *t*-test with Bonferroni correction for multiple comparisons. ** p<0.01; *** p<0.001; n.s, not significant. Scale bar is 20 μm.

compared to either single mutant (***Figure 8F***). This outcome is similar to what we observed in the *muIs32* background (***Figure 8D***). Finally, when comparing the incidence of PLM hook defects in *vab-1; rpm-1* double mutants to single mutants in the *zdIs5* background, we did not observe any significant differences (***Figure 8D***).

Collectively, these results suggest two general conclusions: (1) The enhanced incidence of axon termination defects in *vab-1; glo-4* and *vab-1; fsn-1* double mutants compared to single mutants indicates that VAB-1/EPHR functions in parallel to known RPM-1 binding proteins to facilitate axon termination. (2) *vab-1; rpm-1* double mutants do not display an increased frequency or severity of axon termination defects compared to *rpm-1* single mutants on multiple transgenic backgrounds.

Thus, because we are using null alleles, we conclude that VAB-1/EPHR functions in the same genetic pathway as RPM-1/MYCBP2.

## Discussion

Our MS-based proteomics efforts to identify EPHB2 interacting proteins yielded MYCBP2, a signalling hub and ubiquitin ligase that is functionally linked to many of the cellular processes also mediated by EPHB2, including cellular growth, proliferation, synapse formation, and axon development. Our experiments argue against EPHB2 being a MYCBP2 ubiquitination substrate. Instead, we envisage a model where MYCBP2 controls EPHB2 signalling indirectly by preventing its lysosomal degradation and maintaining EPHB2 protein levels sufficiently high to mediate efficient cellular and axonal growth cone repulsion from ephrin-B ligands. The interaction between MYCBP2 and EPHB2 may allow the coupling EPHB2 to fundamental cellular processes that control growth, proliferation, and survival. Here, we discuss the molecular logic of MYCBP2 and EPHB2 association in the context of Eph receptor signalling, its potential implications for neural development and diversification of EphB2 signalling.

### Functional significance of MYCBP2-EPHB2 complex formation

Our biochemical experiments validate the formation of a MYCBP2-EPHB2 complex and suggest that this association is decreased following ligand application. Because PHR proteins like MYCBP2 are large signalling hubs, MYCBP2 association with EPHB2 might sterically hinder the formation of EPHB2 multimers and clusters necessary for signalling. Thus, ligand-induced dissociation of the MYCBP2-EPHB2 complex could be a prelude to signalling. Our findings indicate that MYCBP2 association with EPHB2 is enhanced by FBXO45, a subunit of the MYCBP2/FBXO45 complex that mediates ubiquitination substrate binding. Previous work showed that MYCBP2 functions to polyubiquitinate specific protein substrates, targeting them for degradation and inhibition (*Crawley et al., 2019*; *Han et al., 2012*; *Nakata et al., 2005*; *Desbois et al., 2022*). We considered the possibility that the ubiquitin ligase activity of MYCBP2 is important for the termination of EPHB2 signalling, but several lines of evidence argue against this idea: (1) the application of ephrin-B2 ligand results in the dissociation of the MYCBP2-EPHB2 complex, (2) loss of MYCBP2 function results in decreased cellular levels of EPHB2 protein, and (3) impairing MYCBP2 increases ligand-stimulated EPHB2 ubiquitination. In addition, our quantification of EPHB2 signalling suggested that decreased EPHB2 and ERK phosphorylation seen in MYCBP2-deficient cells can be accounted for by the decrease in EPHB2 receptor levels. Thus, a more plausible model that is consistent with our results is that MYCBP2 association with EPHB2 protects EPHB2 from turnover by lysosome-mediated degradation and prevents EPHB2 ubiquitination by the action of unidentified ubiquitin ligases. Interestingly, studies in *C. elegans* have shown that RPM-1/MYCBP2 regulates lysosome biogenesis via the GLO-4/GLO-1 pathway (*Grill et al., 2007*). While this established a link between MYCBP2 signalling and the endo-lysosomal degradation system, our results now indicate that MYCBP2 can also influence turnover of EPHB2 via lysosomal degradation. Our *C. elegans* experiments reveal complex genetic interactions between VAB-1/EPHR and members of the RPM-1/MYCBP2 signalling network. These are consistent with VAB-1/EPHR and RPM-1/MYCBP2 acting in the same pathway, while suggesting that VAB-1/EPHR acts in a parallel genetic pathway with GLO-4, the RPM-1 binding protein that mediates effects on lysosome biogenesis. As a caveat, we note that because *C. elegans* contains only a single Eph receptor, our findings with do not necessarily pertain specifically to EPHB2.

The 'protective' effect of MYCBP2 vis-à-vis EPHB2 ubiquitination and lysosomal degradation might be secondary to effects on other aspects of EPHB2 signalling that we have not explored experimentally. For example, since MYCBP2 is a signalling hub with multiple substrates and binding proteins, it could bring EphB2 into close proximity to components of the MYCBP2 signalling network integrating cell-cell communication via ephrin:Eph signals with MYCBP2 intracellular signalling to influence fundamental cellular processes. One example of this is suggested by a recent study demonstrating a link between EPHB2 and mTOR-mediated cell growth signalling pathways (*M Gagné et al., 2021*). One potential mechanism could involve Ephrin binding-induced dissociation of the EPHB2-MYCBP2 interaction, allowing MYCBP2 ubiquitination of the TSC complex that regulates mTOR function (*Han et al., 2012*).

The formation of a MYCBP2-EPHB2 complex that includes other signalling receptors could also explain the curious result that the extracellular domain of EPHB2 is critical for MYCBP2 association. The finding would be consistent with the extracellular domain of EPHB2 interacting with the extracellular domains of other receptors, whose intracellular domains are linked more directly with MYCBP2 and FBXO45. An alternative explanation may be a non-classical extracellular MYCBP2-EPHB2 interaction similar to the one proposed for the extracellular domain of N-Cadherin and FBXO45 (*Na et al., 2020*).

## MYCBP2 and EPHB2 functions in the developing nervous system

Our proteomic, biochemical, and genetic experiments indicate that MYCBP2 and EPHB2 function in the same pathway. This is also supported by the striking similarity of MYCBP2 and EPHB2 loss-of-function phenotypes in the mouse nervous system. In the context of developing neuronal connections, *EphB2* and *Mycbp2/Phr1* mouse mutants display similar axon guidance phenotypes such as abnormal limb nerve trajectories by motor axons, defective growth cone crossing of the midline at the level of the optic chiasm and decreased connectivity between the two cortical hemispheres through the corpus callosum (*Lewcock et al., 2007*; *Luria et al., 2008*; *Williams et al., 2003*; *Henkemeyer et al., 1996*; *D'Souza et al., 2005*). Both proteins also function in synaptic development and their loss of function leads to decreased numbers of synapses and altered synapse morphology (*Wan et al., 2000*; *Bloom et al., 2007*; *Dalva et al., 2000*; *Zhen et al., 2000*). Furthermore, outside the nervous system, both are involved in a variety of cancers with recent evidence linking them to esophageal adenocarcinoma and c-MYC-dependent control of cell proliferation (*Venkitachalam et al., 2022*; *Han et al., 2012*; *Genander et al., 2009*). Our proteomic, biochemical, and cellular experiments together with genetic interaction studies in *C. elegans* now provide a new framework in which to consider phenotypic and disease links between EPHB2 and MYCBP2.

Importantly, the biomedical relevance of our findings are heightened by a recent study that identified genetic variants in MYCBP2, which cause a neurodevelopmental disorder termed *MYCBP2*-related Developmental delay with Corpus callosum Defects (MDCD) (*AlAbdi et al., 2023*). MDCD features defective neuronal connectivity including a hypoplastic or absent corpus callosum, neurobehavioral deficits including intellectual disability and epilepsy, and abnormal craniofacial development. This constellation of comorbidities in MDCD closely resembles some of the phenotypes observed in mice with deficient EPHB2 signalling. Given our finding that MYCBP2 loss reduces EPHB2 levels and influences Eph receptor effects on axon development, the question of whether EPHB2 expression levels are normal in MYCBP2 patients remains pertinent.

In conclusion, our study has revealed numerous biochemical and genetic links between MYCBP2 and EPHB2. Our findings indicate that the MYCBP2/FBXO45 complex protects EPHB2 from degradation, and these are functionally integrated signalling players with an evolutionarily conserved role in axonal development. Future studies will be needed to address how the EPHB2-MYCBP2 interaction affects nervous system development in mammals in vivo and to identify further regulators of EPHB2 degradation. Additionally, another idea worthy of closer examination in the future is the possibility that MYCBP2 signalling could provide routes through which EPHB2-initiated signals access numerous fundamental cellular functions.

# Materials and methods

**Key resources table**

| Reagent type (species) or resource | Designation | Source or reference | Identifiers | Additional information |
|---|---|---|---|---|
| Strain, strain background (*Mus musculus*) | C57BL/6 J | Jackson Labs | Jax:000664 | |
| Cell line (*Homo sapiens*) | HeLa | ATCC | #CCL2 | |
| Cell line (*Homo sapiens*) | HEK 293T | ATCC | #CRL-3216 | |
| Cell line (*Homo sapiens*) | T-REx-HeLa | Invitrogen | #R71407 | |

*Continued on next page*

*Continued*

| Reagent type (species) or resource | Designation | Source or reference | Identifiers | Additional information |
|---|---|---|---|---|
| Cell line (*Homo sapiens*) | HeLa CTRL CRISPR | This paper | | Materials and methods: Cell culture |
| Cell line (*Homo sapiens*) | HeLa *MYCBP2* CRISPR | This paper | | Materials and methods: Cell culture |
| Antibody | Anti-MYCBP2 (Rabbit polyclonal) | Abcam | RRID:AB_1925230 | WB:(1:2000) |
| Antibody | Anti-pERK1/2 (Rabbit polyclonal) | Cell Signaling Technology | RRID:AB_331646 | WB:(1:1000) |
| Antibody | Anti-ERK1/2 (Rabbit polyclonal) | Cell Signaling Technology | RRID:AB_330744 | WB:(1:1000) |
| Antibody | Anti-EPHB2 (Goat polyclonal) | R&D Systems | RRID:AB_355375 | WB:(1:1000) |
| Antibody | Anti-β-Actin (Mouse monoclonal) | Sigma-Aldrich | RRID:AB_476744 | WB:(1:4000) |
| Antibody | Anti-pTyr (Mouse monoclonal) | Santa Cruz Biotechnology | RRID:AB_628122 | WB:(1:400) |
| Antibody | Anti-GAPDH (Mouse monoclonal) | Santa Cruz Biotechnology | RRID:AB_627678 | WB:(1:1000) |
| Antibody | Anti-FLAG (Mouse monoclonal) | Sigma-Aldrich | RRID:AB_439702 | WB:(1:2000) |
| Antibody | Anti-HA (Mouse monoclonal) | Cell Signaling Technology | RRID:AB_1549585 | WB:(1:1000) |
| Antibody | Anti-MYC (Mouse monoclonal) | Santa Cruz Biotechnology | RRID:AB_627268 | WB:(1:400) |
| Antibody | anti-GFP (Rabbit polyclonal) | Thermo Fisher Scientific | RRID:AB_221569 | WB:(1:1000) |
| Antibody | Donkey anti- Goat HRP | Jackson ImmunoResearch | #705-035-003 | WB:(1:5000) |
| Antibody | Donkey anti-Mouse HRP | Jackson ImmunoResearch | #715-035-151 | WB:(1:5000) |
| Antibody | Donkey anti-Rabbit HRP | Jackson ImmunoResearch | #711-035-152 | WB:(1:5000) |
| Antibody | Goat anti-Fc IgG | Sigma-Aldrich | #I2136 | for conjugation |
| Peptide, recombinant protein | Fc | Millipore | #401104 | |
| Peptide, recombinant protein | ephrinB1-Fc | R&D Systems | #473-EB | |
| Peptide, recombinant protein | ephrinB2-Fc | R&D Systems | #496-EB | |
| Chemical compound, drug | Penicillin/Streptomycin | Wisent Bioproducts | #450–200-EL | for cell line culture |
| Chemical compound, drug | Penicillin/Streptomycin | Hyclone | #SV30010 | for neuron culture |
| Chemical compound, drug | MG132 | Sigma-Aldrich | #474790 | 50 µM |
| Chemical compound, drug | BafilomycinA1 | Sigma-Aldrich | #B1793 | 0.2 µM |
| Chemical compound, drug | Chloroquine | Tocris | #4109 | 50 µM |

*Continued on next page*

*Continued*

| Reagent type (species) or resource | Designation | Source or reference | Identifiers | Additional information |
|---|---|---|---|---|
| Other | Neurobasal | Thermo Fisher Scientific | #21103049 | |
| Other | B-27 | Thermo Fisher Scientific | #17504044 | |
| Other | HBSS | Gibco | #14185052 | Materials and methods: Dissociated mouse hippocampal neuron culture and electroporation |
| Other | HEPES | Gibco | #15630080 | |
| Other | SM1 supplement | Stemcell | #05711 | |
| Other | GlutaMAX-I | Gibco | #35050 | |
| Other | Protease inhibitor | Roche | #11836153001 | |
| Other | Phosphatase inhibitor | Roche | #04906837001 | Materials and methods: Cell lysis, co-immunoprecipitation and western blotting |
| Other | Anti-FLAG agarose beads | Sigma-Aldrich | #A2220 | |
| Other | PVDF membrane | Millipore | #IPVH00010 | |

## Vertebrate animals

All animal experiments were carried out in accordance with the Canadian Council on Animal Care guidelines and approved by the IRCM Animal Care Committee (Protocol 2019–09 AK and 2021–12 AK). Fertilized chicken eggs (FERME GMS, Saint-Liboire, QC, Canada) were incubated at 38–39°C and staged according to Hamburger and Hamilton (HH) (*Hamburger and Hamilton, 1951*). C57BL/6 mice were used for hippocampal neuron collapse assay. Timed mating vaginal plug was designated as E0.5.

## Cell culture

HeLa and HEK293T cells were maintained in DMEM (Thermo Fisher Scientific, #11965092) supplemented with 10% Fetal Bovine Serum (FBS; Wisent Bioproducts, #080–150) and 1% Penicillin/ Streptomycin (Wisent Bioproducts, #450–200-EL) at 37 °C with 5% $CO_2$. Tetracycline inducible HeLa EPHB2-FLAG cells were generated by transfecting Flp-In T-REx HeLa cells with EPHB2-BirA*-FLAG expression plasmid using Lipofectamine 3000 (Invitrogen, #L3000015) followed by hygromycin selection (200 µg/ml). Stable but not clonal CTRL$^{CTRISPR}$ or MYCBP2$^{CRISPR}$ HeLa EPHB2-FLAG cell lines were generated by infecting cells with packaged lentivirus, followed by puromycin selection (1 µg/ ml). Lentivirus particles were packaged using MYCBP2 sgRNA CRISPR plasmid designed to target the *MYCBP2* exon6 (pLentiCRISPR2-sgMYCBP2) and pLentiCRISPR empty vector was used as Ctrl CRISPR. EPHB2-FLAG overexpression was induced using 1 µg/ml tetracycline simultaneously with cell starvation in DMEM supplemented with 0.5%FBS, 1% penicillin/streptomycin for 12–20 hr. Prior to cell stimulation, Fc control (Millipore, #401104), ephrinB1-Fc (R&D, #473-EB) or ephrinB2-Fc (R&D, #496-EB) were pre-clustered using goat anti-human Fc IgG (Sigma, #I2136) in 4:1 ratio for 30 min.

## Affinity purification - mass spectrometry

HeLa EPHB2-FLAG cells cultured in DMEM supplemented with 0.5% FBS and 1 µg/ml tetracycline in 15 cm cell culture plates for 20 hr, were treated with pre-clustered Fc control or ephrinB2-Fc for 15 min. After treatment, cells were washed twice with PBS and lysed using a lysis buffer (50 mM Tris, pH7.4; 150 mM NaCl; 1% NP-40) supplemented with protease (Roche, #11836153001) and phosphatase inhibitors (Roche, #04906837001). The lysates were collected in 1.5 ml Eppendorf tubes and centrifuged at 13,000 rpm for 15 min at 4 °C. Supernatants were transferred to new tubes with prewashed anti-FLAG agarose beads (Sigma, #A2220) and incubated on a rotator overnight at 4 °C. The following day, beads were washed four times using 50 mM Ammonium Bicarbonate. The on-bead proteins were diluted in 2 M Urea/50 mM ammonium bicarbonate and on-bead trypsin digestion was performed overnight at 37 °C. The samples were then reduced with 13 mM dithiothreitol at 37 °C and, after cooling for 10 min, alkylated with 23 mM iodoacetamide at room temperature for 20 min in the dark. The supernatants were acidified with trifluoroacetic acid and cleaned from residual detergents and reagents with MCX cartridges (Waters Oasis MCX 96-well Elution Plate) following the manufacturer's instructions. After elution in 10% ammonium hydroxide /90% methanol (v/v), samples were dried with a Speed-vac, reconstituted under agitation for 15 min in 12 µL of 2%ACN-1%FA and loaded into

a 75 µm i.d. ×150 mm Self-Pack C18 column installed in the Easy-nLC II system (Proxeon Biosystems). Peptides were eluted with a two-slope gradient at a flowrate of 250 nL/min. Solvent B first increased from 2 to 35% in 100 min and then from 35 to 80% B in 10 min. The HPLC system was coupled to Orbitrap Fusion mass spectrometer (Thermo Scientific) through a Nanospray Flex Ion Source. Nanospray and S-lens voltages were set to 1.3–1.7 kV and 60 V, respectively. Capillary temperature was set to 225 °C. Full scan MS survey spectra (m/z 360–1560) in profile mode were acquired in the Orbitrap with a resolution of 120,000 with a target value at 3e5. The 25 most intense peptide ions were fragmented in the HCD collision cell and analyzed in the linear ion trap with a target value at 2e4 and a normalised collision energy at 29 V. Target ions selected for fragmentation were dynamically excluded for 15 s after two MS2 events.

## Mass spectrometry data analysis

The peak list files were generated with Proteome Discoverer (version 2.3) using the following parameters: minimum mass set to 500 Da, maximum mass set to 6000 Da, no grouping of MS/MS spectra, precursor charge set to auto, and minimum number of fragment ions set to 5. Protein database searching was performed with Mascot 2.6 (Matrix Science) against the UniProt Human protein database. The mass tolerances for precursor and fragment ions were set to 10 ppm and 0.6 Da, respectively. Trypsin was used as the enzyme allowing for up to 1 missed cleavage. Cysteine carbamidomethylation was specified as a fixed modification, and methionine oxidation as variable modification. Data interpretation was performed using Scaffold (version 4.8) and further statistical analysis was performed through ProHits integrated with SAINT (Significance Analysis of INTeractome) (*Liu et al., 2010*).

## Cell lysis, co-immunoprecipitation and western blotting

Cells were washed with PBS, lysed with RIPA buffer (50 mM Tris, pH 7.4; 150 mM NaCl; 1% NP-40; 0.1% SDS) supplemented with protease and phosphatase inhibitors. For co-IP experiments, cells were lysed with co-IP buffer (50 mM Tris, pH 7.4; 150 mM NaCl; 0.1% NP-40) with protease and phosphatase inhibitors. Cell lysates were centrifuged at 13,000 rpm for 20 min at 4 °C, then the supernatants were collected, and total protein concentrations were quantified using BCA kit (Thermo Fisher Scientific, #23225). For FLAG co-IP, 500–1000 µg of total protein was incubated with 20–40 µl of prewashed anti-FLAG agarose beads (Sigma, #A2220) for 3 hr at 4 °C. After incubation, the beads were centrifuged at 2600 rpm for 1 min at 4°C and washed three times with the co-IP buffer. The beads were resuspended in 2xSDS-PAGE loading buffer (5 x loading buffer: Tris, 150 mM, pH 6.8; SDS, 10%; Glyercol, 30%; b-Mercatoethanol, 5%; Bromophenol Blue, 0.02%). For western blotting, proteins were separated on 6–10% SDS-PAGE gels and transferred to methanol pre-activated PVDF membranes (Millipore, #IPVH00010). For MYCBP2 blots, gels were wet transferred overnight at 4 °C using 33 V. Membranes were incubated in blocking buffer (TBST: 20 mM Tris, pH 7.6; 150 mM NaCl; 0.1% Tween 20; 5% skim milk) for 1 hr at room temperature, followed by primary antibody incubation (1–2 hr at room temperature or overnight at 4 °C) and corresponding secondary antibody incubation (1 hr at room temperature). Primary antibodies were: rabbit polyclonal anti-MYCBP2 (Abcam, #ab86078; RRID:AB_1925230), rabbit anti-pERK1/2 (Thr202/Tyr204; Cell Signaling Technology, #9101; RRID:AB_331646), rabbit anti-ERK1/2 (Cell Signaling Technology, #9102; RRID:AB_330744), goat polyclonal anti-EPHB2 (R&D Systems, #AF467; RRID:AB_355375), mouse monoclonal anti-Actin (Sigma-Aldrich, #A5441; RRID:AB_476744), mouse monoclonal anti-pTyr (PY20; Santa Cruz Biotechnology, #sc-508; RRID:AB_628122), mouse monoclonal anti-GAPDH (Santa Cruz Biotechnology, #sc-47724; RRID:AB_627678), mouse monoclonal anti-FLAG-HRP (Sigma-Aldrich, #A8592; RRID:AB_439702), rabbit monoclonal anti-HA (Cell Signaling Technology, #3724; RRID:AB_1549585), mouse monoclonal anti-MYC (Santa Cruz Biotechnology, #sc-40; RRID:AB_627268), rabbit polyclonal anti-GFP (Thermo Fisher Scientific, #A-11122; RRID:AB_221569). Secondary antibodies were: Donkey anti-Goat HRP (Jackson, 705-035-003), Donkey anti-Mouse HRP (Jackson, 715-035-151), Donkey anti-Rabbit HRP (Jackson, 711-035-152). After three washes with TBST, membranes were incubated with ECL reagent (Cytiva, RPN2106) for 1 min and chemiluminescence signal was acquired using film or Bio-Rad ChemiDoc Imaging machine. Band intensity was quantified using ImageJ or Bio-Rad Image Lab software.

## HeLa cell collapse assay

HeLa-EPHB2 CTRL<sup>CRISPR</sup> and MYCBP2<sup>CRISPR</sup> cells were seeded on glass coverslips (Electron Microscopy Sciences, #7223101) in 24-well plates at a density of 20,000 cells/well. After 24 hr, cell media was changed for DMEM supplemented with 0.5% FBS, 1% P/S and 1 µg/ml tetracycline to starve the cells and induce EPHB2 expression for 16–20 hr. Cells were then stimulated with 1.5 µg/ml of pre-clustered Fc control or ephrin-B2-Fc for 15 min. Cells were fixed with 3.2% paraformaldehyde (Lewcock et al.), 6% sucrose in PBS for 12–15 min. Nuclei were stained with DAPI and F-actin with Phalloidin Alexa Fluor 568 conjugate (Thermo Fisher, #A12380). Images were acquired using Zeiss LSM710 confocal microscope and 20 x objective. Fully rounded cells are scored as collapsed cells.

For time lapse imaging experiments, CTRL<sup>CRISPR</sup> or *MYCBP2*<sup>CRISPR</sup> cells were plated on Poly-D-Lysine coated glass bottom 35 mm dishes (MATTEK, #P35GC-1.5–10 C) at a density of 300,000 cells/dish. The next day, cells were transfected with 1.5 µg of EPHB2-GFP plasmid for 4–5 hr, using lipofectamine 3000 in opti-MEM (ThermoFisher, #31985070), and then media was changed for DMEM supplemented with 0.5% FBS. The following day, the images were acquired under Zeiss Spinning Disk Microscope using a 20 x objective. During the imaging, the cells were maintained at 37 °C with 5% $CO_2$. Pre-clustered ephrinB2-Fc was added to a final concentration of 2 µg/mL directly into the dishes at the beginning of each experiment. The images were acquired every minute for 1 hr.

## HeLa cell stripe assay

Alternative ephrin-B2-Fc or Fc stripes were prepared using silicon matrices with a micro-well system (*Poliak et al., 2015*). HeLa EPHB2 CTRL<sup>CRISPR</sup> and MYCBP2<sup>CRISPR</sup> cells, or HeLa EPHB2 cells transiently transfected with GFP-FBD1 wild-type or GFP-FBD1 mutant (GRR/AAA: G2404A, R2406A, R2408A), were cultured with tetracycline for 20 hr, trypsinized and plated on stripes (~10,000 cells per carpet); see *Desbois et al., 2018* for specific sequences. The next day, cells were fixed with paraformaldehyde (3.2% PFA, 6% sucrose in PBS), and stained with DAPI and Phalloidin iFluor 488 (Abcam, #ab176753); or DAPI, Rabbit anti-GFP (Thermo Fisher Scientific, # A-11122; RRID:AB_221569) and Phalloidin iFluor 647 (Abcam, #ab176759). Images were acquired using Zeiss LSM710 confocal microscope and 20 x objective (three vision fields for each carpet). A cell was considered to be on an ephrin-B2 stripe when more than 50% of its nucleus was located on that stripe.

## Ubiquitination assay

HeLa EPHB2 CTRL<sup>CRISPR</sup> and MYCBP2<sup>CRISPR</sup> cells were seeded in six-well plates at a density of 0.5 million cells per well. Next day, cells were transfected with 1.2 µg HA-Ubiquitin (gift of Gu Hua) using lipofectamine 3000 (2µl/well) for 4–5 hr in opti-MEM (ThermoFisher, #31985070), then media was changed to DMEM with 10%FBS. The following day, EPHB2 expression was induced using 1 µg/ml tetracycline in DMEM with 0.5% FBS for 12 hr followed by 2 µg/ml Fc or eB2-Fc treatment for 30 min. After IP using anti-FLAG beads, precipitates were eluted with 2xSDS loading buffer, resolved using 8% SDS gel and transferred onto PVDF membranes. Membranes were blocked in 5% milk following an incubation with anti-HA antibody (Cell Signaling Technology, #3724) to detect EPHB2 ubiquitination levels. The membrane was then stripped using mild stripping buffer (1 L: 15 g glycine, 1 g SDS, 10 mL Tween 20, pH 2.2) and probed with anti-FLAG antibody to reveal EPHB2 levels.

## Lysosome and proteasome inhibition

HeLa EPHB2 CTRL<sup>CRISPR</sup> and MYCBP2<sup>CRISPR</sup> cells were seed in 6-well plates at a density of 0.5 million cells per well. Next day, EPHB2 overexpression was induced using 1 µg/ml tetracycline in DMEM with 10% FBS for 16 hr, followed by 26 S proteasome inhibitor (MG132, 50 µM, Sigma, #474790) or lysosome inhibitor treatment (BafilomycinA1, 0.2 µM, Sigma, #B1793; Chloroquine, 50 µM, Tocris, #4109) for 6 hr.

## Chick in ovo electroporation

Fertilized eggs (FERME GMS, Saint-Liboire, QC) were incubated in an incubator (Lyon Technologies, model PRFWD) at 39 °C with a humidity level of around 40%–60% according to standard protocols. At HH stage.15–17, chick embryo spinal neural tubes were electroporated with expression constructs (TSS20 Ovodyne electroporator at 30 V, 5 pulses, 50ms wide, 1000ms interval). Following

electroporation, eggs were sealed with double layer of parafilm (Pechiney Plastic Packaging Company) and incubated till HH stage 24–26.

## Chick spinal explants stripe assay

Alternative eprhinB2-Fc/Fc stripes were prepared using silicon matrices with a micro-well system and pre-coated with laminin (*Poliak et al., 2015*). At HH stage 24–26, chick embryos were harvested, and the lumbar part neural tubes were dissected with tungsten needles (World Precision Instruments) in MN medium (20 ml motor neuron medium: 19.2 ml Neurobasal medium, 400 µl B27 supplement, 200 µl 50 mM L-glutamic acid, 200 µl 100xP/S antibiotics,73mg L-glutamine). The lumbar neural tube was then cut into around 20 explants which were plated on stripes (a 1cmx1cm square covers the whole stripe area). After overnight incubation, the explants were fixed with 4% PFA for 12 min at 37 °C, washed once with PBS, and incubated with blocking buffer, primary antibodies, and secondary antibodies. After three PBS washes, the samples were mounted and neurites extending from explants were imaged using LSM710. The fraction of GFP signal on ephrin-B2 stripes was calculated by measuring the total length of GFP-expressing neurites found on ephrin-B2 stripes divided by the total length of GFP-expressing neurites found on either stripe. The number of explants with significant outgrowth varied between one and five per stripe.

## Dissociated mouse hippocampal neuron culture and electroporation

Primary hippocampal neurons were cultured from wild-type C57BL/6 mice at embryonic day 16–18 (E16-18). The hippocampi were dissected out and collected in 4.5 ml dissection buffer (calcium- and magnesium-free Hank's BSS: 500 ml distilled water (Gibco, #15230162), 56.8 ml 10xHBSS (Gibco, #14185052), 5.68 ml 1 M HEPES (Gibco, #15630080), 2.84 ml HyClone (Thermo Scientific, #SV30010)). Hippocampi were added with 0.5 ml 2.5%Tyrpsin and incubated at 37 °C for 13–15 min. After five times of thorough wash with dissection buffer, hippocampal neurons were dissociated in 0.8 ml DMEM (Gibco, #11965118) with 10% FBS (Wisent Bioproducts, 080–150) by pipetting 10 times up and down. Then cell numbers were counted and desired number of neurons were directed for electroporation. After spin down at 2000 rpm for 2 min, dissociated hippocampal neurons (1x10$^6$/condition) were resuspended with 100 µl homemade nucleofection solution, mixed with 5 µg of DNA, and transferred into the aluminum cuvettes (AMAXA/Lonza). Electroporation was achieved by Nucleofector I (AMAXA/Lonza) using program O-05 (Mouse CNS neurons). 1 ml Plating Medium (PM; 500 ml MEM (Sigma, #M4655), 17.5 ml 20% Glucose (Sigma, #G8270), 5.8 ml 100 mM pyruvate (Sigma, #P2256), 58 ml heat-inactivated horse serum (Thermo Scientific, #26050088)) was added to the cuvettes immediately, and desired number of neurons were plated on 1 mg/ml Poly-L-Lysine (Sigma, #P2636) coated coverslips in 12-well plate. At 1 day in vitro (DIV1), medium was replaced with Neuron growth and maintenance medium (NBG; 500 ml Neurobasal medium (Gibco, #21103049), 10 ml SM1 neuronal supplement (Stemcell, #05711), and 1.25 ml GlutaMAX-I (Gibco, #35050)).

## Hippocampal neuron growth cone collapse assay

Electroporated hippocampal neurons were cultured on glass coverslips (18 mm, 100 thousand neurons/well in a 12-well plate). At DIV2, the neurons were treated with pre-clustered ephrinB1-Fc or Fc control in NBG medium at a final concentration of 2 µg/ml for 60 min at 37 °C. After treatment, neurons were fixed with paraformaldehyde (3.2% PFA,6% sucrose in PBS) for 12 min, followed by two PBS washes, and blocked with Blocking Buffer (PBS, 0.15% TritonX-100, 2% FBS) for 60 min at room temperature. Neurons were then incubated with rabbit anti-GFP antibody (1:5000, Thermo Fisher Scientific, #A11122; RRID:AB_221569) in Blocking Buffer for 90 min at room temperature. Followed by three PBS washes, neurons were incubated with DAPI, donkey anti-rabbit IgG Alexa Fluor 488 conjugate (1:1000, Jacksonimmuno, #711545152) and Phalloidin Alexa Fluor 568 conjugate (1:300, Invitrogen, #A12380) in Blocking Buffer for 60 min. Neurons on coverslips were then washed and mounted on microscope slides (Fisherbrand, #1255015). Collapsed growth cones were scored using followed criteria:

## Collapsed hippocampal neuron growth cone quantification

Neuron selection: only neurons with moderate GFP expression and only neurons with more than three neurites were scored (neurons with strong GFP expression were excluded). Neurite selection: neurites

shorter than the diameter of the neuron cell body were excluded; neurites intermingling with others were excluded. Collapsed growth cone: growth cones with a fan shape were scored as full, growth cones with a width smaller than that of the neurite were scored as collapsed, and growth cones' size in between were scored as 'hard to tell'. The collapse rate was calculated using collapse growth cone numbers divided by the total growth cone numbers.

## Microscopy and imaging

HeLa cell collapse assay images were acquired using Leica DM6 or Zeiss LSM710 confocal microscopy. Stripe assay and growth cone collapse assay images were acquired with a Zeiss LSM710 or LSM700 confocal microscope.

## Quantification and statistical analysis

All cell counts, collapsed/uncollapsed visualization and explant neurite length measurements were performed with ImageJ 2.9.0 (*Schindelin et al., 2012*). All numbers are illustrated in figure legends. In western blotting, each n represents one independent experiment; in neuron growth cone collapse assay, each n represents one independent experiment with neurons pooled from multiple embryos. All data statistical analyses were performed using GraphPad Prism 9.1.1. Test methods and p values were described in figure legends, with p value 0.05 as a significance threshold.

## *C. elegans* genetics and strains

*C. elegans* N2 isolate was used for all experiments. Animals were maintained using standard procedures. The following mutant alleles were used: *vab-1(dx31)* II, *fsn-1(gk429)* III, *rpm-1(ju44)* V, *glo-4(ok623)* V. All mutant alleles are likely genetic or protein nulls. The integrated transgenes used to evaluate axon termination were *muIs32* [$P_{mec-7}$GFP] II and *zdIs5* [$P_{mec-4}$GFP] I. For genetic analysis the animals were grown at 23 °C.

Transgenic extrachromosomal arrays were generated using standard microinjection procedures for *C. elegans. vab-1* minigene (pCZ47) was injected at either 25 ng/μL or 50 ng/μL, and co-injection markers used for transgene selection were either neomycin resistance (pBG-264) or P*ttx-3*::RFP (pBG-41). pBluescript (pBG-49) was used to reach a final concentration of 100 ng/μl in all injection mixes. pCZ47 was a gift from Andrew Chisholm (Addgene plasmid # 128414; RRID:Addgene_128414).

## *C. elegans* axon termination analysis and imaging

Axon termination defects were defined as PLM axons that extended beyond the normal termination point adjacent to the ALM cell body. Two different failed termination phenotypes were scored: axon overextension (moderate phenotype) where the PLM axon grew beyond ALM cell body, axons that overextend and form a ventral hook (severe phenotype). To quantify axon termination defects, 20–30 young adult animals were anesthetized (10 μM levamisole in M9 buffer) on a 2% agar pad on glass slides and visualized with a Leica DM5000 B (CTR5000) epifluorescent microscope (40 x oil-immersion objective). For image acquisition, young adult animals were mounted on a 3% agarose pad and a Zeiss LSM 710 (40x oil-immersion objective) was used to generate z stacks.

For statistical analysis of axon termination defects, comparisons were done using a Student's *t*-test with Bonferroni correction for multiple comparisons on GraphPad Prism software. Error bars represent standard error of the mean (SEM). Significance was defined as p<0.05 after Bonferroni correction. Bar graphs represent averages from 5 to 10 counts (20–30 animals/count) obtained from five or more independent experiments for each genotype. For transgenic rescue experiments, data shown in *Figure 8B* was obtained from two, independently derived transgenic extrachromosomal arrays.

## Acknowledgements

We thank Dominic Fillion for assistance with microscopy, Denis Flaubert and the IRCM Proteomics Core for proteomics analysis, Gu Hua for HA-Ubiquitin plasmid, Sylvie Lahaie, Louis-Philippe Croteau and Farin B Bourojeni for discussions, and Meirong Liang for technical support. This work was supported by project grants from the Canadian Institutes of Health Research (PJT-162225, MOP-77556, PJT-153053, and PJT-159839) to AK and a grant from the National Institutes of Health (R01 NS072129) to BG. CC received IRCM Jacques-Gauthier Scholarship. SLB is a Fonds de recherche du Québec – Santé (FRQS) scholar. SSP is funded by IRCM Jean-Coutu Scholarship.

## Additional information

### Funding

| Funder | Grant reference number | Author |
| --- | --- | --- |
| Canadian Institutes of Health Research | PJT-162225 | Artur Kania |
| National Institutes of Health | R01 NS072129 | Brock Grill |
| Canadian Institutes of Health Research | PJT-153053 | Artur Kania |
| Canadian Institutes of Health Research | PJT-159839 | Artur Kania |
| Canadian Institutes of Health Research | MOP-77556 | Artur Kania |

The funders had no role in study design, data collection and interpretation, or the decision to submit the work for publication.

### Author contributions

Chao Chang, Conceptualization, Data curation, Investigation, Methodology, Visualization, Writing – original draft, Writing – review and editing; Sara L Banerjee, Conceptualization, Data curation, Investigation, Methodology, Supervision, Visualization, Writing – original draft, Writing – review and editing; Sung Soon Park, Conceptualization, Data curation, Investigation, Methodology, Visualization, Writing – review and editing; Xiao Lei Zhang, David Cotnoir-White, Conceptualization, Investigation, Writing – review and editing, Writing – original draft; Karla J Opperman, Conceptualization, Data curation, Investigation, Writing – review and editing, Writing – original draft; Muriel Desbois, Conceptualization, Data curation, Investigation, Methodology, Resources, Supervision, Visualization, Writing – original draft, Writing – review and editing; Brock Grill, Conceptualization, Supervision, Funding acquisition, Visualization, Writing – original draft; Artur Kania, Conceptualization, Supervision, Funding acquisition, Visualization, Project administration, Writing – original draft

### Author ORCIDs

Chao Chang ⓘ http://orcid.org/0009-0007-3154-996X
Sara L Banerjee ⓘ http://orcid.org/0000-0003-0295-2825
Muriel Desbois ⓘ https://orcid.org/0000-0002-8070-2525
Artur Kania ⓘ https://orcid.org/0000-0002-5209-2520

### Ethics

All vertebrate animal experiments were carried out in accordance with the Canadian Council on Animal Care guidelines and approved by the IRCM Animal Care Committee (Protocol 2019-09 AK and 2021-12 AK).

Reviewer #1 (Public Review): https://doi.org/10.7554/eLife.89176.4.sa1
Reviewer #2 (Public Review): https://doi.org/10.7554/eLife.89176.4.sa2
Reviewer #3 (Public Review): https://doi.org/10.7554/eLife.89176.4.sa3
Author Response https://doi.org/10.7554/eLife.89176.4.sa4

## Additional files

### Supplementary files
• MDAR checklist

## Data availability

The mass spectrometry proteomics data have been deposited to the ProteomeXchange Consortium via the PRIDE partner repository (*Perez-Riverol et al., 2019*) with the dataset identifier PXD041786 and https://doi.org/10.6019/PXD041786.

The following dataset was generated:

| Author(s) | Year | Dataset title | Dataset URL | Database and Identifier |
|---|---|---|---|---|
| Chang C, Banerjee SL | 2023 | Ubiquitin ligase and signalling hub MYCBP2 is required for efficient EPHB2 tyrosine kinase receptor function | https://doi.org/10.6019/PXD041786 | ProteomeXchange, 10.6019/PXD041786 |

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
