## [Editor Report · eLife assessment]

This **valuable** study identifies an Ephrin type-B Receptor 2 (EPHB2) interactor, MYCBP2, as a potential regulator of EPHB2 stability and function. In contrast to expectations, based on MYCBP2 function in the ubiquitin pathway, loss of function of MYCBP2 resulted in less EPHB2 receptor and defective EPHB2 function. The paper is supported by a largely **convincing** set of biochemical, cell culture and in vivo experiments.

---

## [Referee Report · Reviewer #1 (Public Review)]

The Eph receptor tyrosine kinase family plays a critical function in multiple physiological and pathophysiological processes. Hence, understating the regulation of these receptors is a highly important question. Through extensive experiments in cell lines and cultured neurons Chang et.al show that the signaling hub protein, MYCBP2 positively regulates the overall stability of a specific member of the family, EPHB2, and by that the cellular response to ephrinBs.

Overall, this work sheds light on the divergent in the regulatory mechanisms of the Eph receptors family. Although the physiological importance of this new regularly mechanism in mammals awaits to be discovered, the authors provide genetic evidence using *C. elegans* that it is evolutionarily conserved.

---

## [Referee Report · Reviewer #2 (Public Review)]

Members of the EphB family of tyrosine kinase receptors are involved in a multitude of diverse cellular functions, ranging from the control of axon growth to angiogenesis and synaptic plasticity. In order to provide these diverse functions, it is expected that these receptors interact in a cell-type specific manner with a diverse variety of downstream signalling molecules.

The authors have used proteomics approaches to characterise some of these molecules in further detail. This molecule, myc-binding protein 2 (MYCBP2) is also known as highwire, has been identified in the context of establishment of neural connectivity. Another molecule coming up on this screen was identified as FBXO45.

The authors use classical methods of co-IP to show a kinase-independent binding of MYCBP2 to EphB2. They further showed that FBXO45 within a ternary complex increased the stability of the EphB2/MYCBP2 complex.

To define the interacting domains, they used clearly designed swapping experiments to show that the extracellular and transmembrane domains are necessary and sufficient for the formation of the ternary complex.

Using a cellular contraction assay, the authors showed the necessity of MYCBP2 in mediating the cytoskeletal response of EphB2 forward signalling. Furthermore, they used the technically challenging stripe assay of alternating lanes of ephrinB-Fc and Fc to show that also in this migration-based essay MYCBP2 is required for EphB mediated differential migration pattern.

MYCBP2 in addition is necessary to stabilize EphB2, that is in the absence of MYCBP2, EphB2 is degraded in the lysosomal pathway.

Interestingly, the third protein in this complex, Fbxo45, was further characterized by overexpression of the domain of MYCBP2, known to interact with Fbxo45. Here the authors showed that this approach led to the disruption of the EphB2 / MYCBP2 complex, and also abolished the ephrinB mediated activation of EphB2 receptors and their differential outgrowth on ephrinB2-Fc / Fc stripes.

Finally, the authors demonstrated an in vivo function of this complex using another model system, C elegans where they were able to show a genetic interaction.

Data show in a nice set of experiments a novel level of EphB2 forward signalling where a ternary complex of this receptor with multifunctional MYCBP2 and Fbxo45 controls the activity of EphB2, allowing a further complex regulation of this important receptors. Additionally, the authors challenge pre-existing concepts of the function of MYCBP2 which might open up novel ways to think about this protein.

Of interest is this work also in terms of development of the retinotectal projection in zebrafish where MYCBP2/highwire plays a crucial role, and thus might lead to a better understanding of patterning along the DV axis, for which it is known that EphB family members are crucial.

Overall, the experiments are classical experiments of co-immunoprecipitations, swapping experiments, collapse assays, and stripe assays which all are well carried out and are convincing.

---

## [Referee Report · Reviewer #3 (Public Review)]

In this improved version of the manuscript, Chang et al set out to find direct interactions with the Eph-B2 receptor, as our knowledge of its function/regulation is still incomplete. Using proteomic analysis of Hela cells expressing EPHB2, they identified MYCBP2 a potential binder, which they then confirm using extensive biochemical analyses, an interaction that seems to be negatively affected by binding of ephrin-B2 (but not B1). Furthermore, they find that FBXO45, a known MYCBP2 interaction, strongly facilitates its binding to EPHB2. Intriguingly, these interactions depend on the extracellular domains of EPHB2, suggesting the involvement of additional proteins as MYCBP2 is thought to be a cytoplasmic protein. Finally, they find that, in contrast to what could be expected given the known function of MYCBP2 as a ubiquitin E3 ligase, it actually positively regulates EPHB2 protein stability, and function.

The strength of this manuscript is the extensive biochemical analysis of the EPHB2/MYCBP2/FBXO43 interactions. The vast majority of the conclusions are supported by the data.

The attempt to extend the study to an in vivo animal using the worm is important, however the additive insight is, unfortunately, minimal.

---

## [Author Response]

The following is the authors’ response to the previous reviews

The revised manuscript is much improved - many unclear points are now better explained. However, in our opinion, some issues could still be significantly improved.1. Statistics: none of us are experts in statistics but several things remain questionable in our opinion and if it were our study, we would consult with an expert:a) while we understand the authors note about N-chasing and p-hacking, we wonder how the number of N's was premeditated before obtaining the results. Why in 4M an N of 3 is sufficient while in 3E the N is >20 (and not mentioned). At the very least, we think it would be wise to be cautious when stating something as not-significant when it is clear (as in 4M) that the likelihood of it actually being statistically significant is quite large.b) In most analyses, the data is not only normalized by actin or some other measure but also to the first (i.e left side on the graph) condition, resulting in identical data points that equal '1' (in Figure 4 alone - C; I; K; M; and O) - while this might be scientifically sound, it should be mentioned (the specific normalization) and also note that this technique shadows any real variance that exists in the original data in this condition. consider exploring techniques to overcome this issue.c) In 3C, - if we understand the experiment, you want to convince us that the DIFFERENCE between eB2-FC compared to FC is larger in the control compared to the experiment. We are not absolutely sure that the statistical tools employed here are sufficient - which is why we would consult an expert.

A) We are aware that many studies do not consistently quantify such experiments. For example, there are essentially no published examples of the signalling timelines of EphB2 receptors as in Fig. 5. By striving to quantifying such biochemical effects, an unquantified experiment stands out, and so perhaps we were too strict by trying to quantify as many experiments as possible, resulting in low n’s for some of them. We acknowledge that additional experiments on EPHB1 protein stability may reach significance. We have adjusted our text on line 332-335 to point to this interesting trend, and slightly changed the conclusion to this section. Similarly, we commented on similar trends when describing Figs. 1E and 4G on lines 901 and 952.

B) For the Western blot band intensity normalisation, we believe that our method is scientifically sound. Normally, when the replicate samples are loaded on one gel and blotted on the same membrane, the experimenter only needs to normalise the target band intensity to its cognate loading control band intensity for quantitation. However, we usually have a large number of samples from multiple experiments, carried out on different dates. For example, in Fig. 4B,C there are 7 biological replicates collected from 7 experiments and in Fig. 4D there are 10 protein samples. It is not possible for us to run all samples on the same gel. In addition, due to the combined effects of variance in transfer efficiency, the potency of antibodies, detection efficiency and the developing time for each blot, it is practically impossible to generate similar band intensity for each batch. Thus, we use normalisation of test bands to the loading control for individual experiments, and this analysis method is widely accepted by reputable journals with a focus on biochemical experiments (for example: PMID 37695914: Fig. 3 A,B,C; PMID 36282215: Fig. 3 B,C,D,E; PMID 33843588: Fig. 3 C,D,E,F,G,H). Since the value of the first sample on the plot is 1, which is a hypothetical value and does not meet the parametric test requirement, we performed one-sample t-test for statistics when other samples are compared with the first sample (PMID 35243233 Fig. 6 A,B,C,D; https://www.graphpad.com/quickcalcs/oneSampleT1/, “A one sample t-test compares the mean with a hypothetical value. In most cases, the hypothetical value comes from theory. For example, if you express your data as 'percent of control', you can test whether the average differs significantly from 100.”). Thus, we believe that our normalisation and statistical methods are both correct with a large number of precedents.

C) This comment refers to the cell collapse experiment shown in Fig. 3C for which the data are plotted in Fig. 3D. We stand by the statistical method used. There are two groups of cells (CTRLCRISPR and MYCBP2 CRISPR) and two treatments for each cell group (Fc control and eB2), thus we should use two-way ANOVA. Since we compared the cell retraction effects of Fc and eB2 on the two groups of cells, Sidak post hoc comparison is the right method to avoid errors introduced by multiple comparisons. Here is an example of an eLife article that used the same statistical method for similar comparisons: PMID 37830910, Fig. 1 H,I. To make the comparison easier, we grouped the experiments by cell type (CTRLCRISPR and MYCBP2 CRISPR) as opposed to by treatment. Below, the old version is on the right, and the new version is on the left. The conclusion is that eB2 induces less cell collapse in cells depleted of MYCBP2, when compared to the control cells. However, eB2 is still able to collapse cells lacking MYCBP2.

**Author response image 1. sa4fig1:** 

Revisiting these data, we noticed an error introduced when CC compiled the data used to generate Fig. 3D. The data were acquired from nine biological replicates per condition. CC used a mix of two methods for cell collapse rate calculation: the first method involved the sum of collapsed cells and all cells from multiple regions of one coverslip (biological replicate). The second method involved computing a collapse rate in each region which then was used to calculate the average collapse rate for the entire coverslip (technical replicate). Given the small cell numbers due to sparse culture conditions, we believe that the first method is a more conservative approach. We hence re-plotted all replicate data using the first method. This resulted in slightly different % collapse and p values. These were changed accordingly in the text and plot and do not affect the conclusion of this experiment.

1. thanks for the clarification that the interaction between the extracellular domain of EPHB2 and MYCBP2 might not occur directly - however, unless we missed this it was not clearly stated in the text. It is an important point and also a cool direction for the future - to find the elusive co-receptor that actually helps EPHB2 and MYCBP2 form a complex.

We now also refer to this in the results section on line 215.

“Since EPHB2 is a transmembrane protein and MYCBP2 is localised in the cytosol, these experiments suggest that the interaction between the extracellular domain of EPHB2 and MYCBP2 might be indirect and mediated by other unknown transmembrane proteins.”

1. The Hela CRISPR cell line is better explained in the response letter but still not sufficiently explained in the text for a non-expert reader. If the authors want any reader to comprehend this, we would strongly recommend adding a scheme.

We now include a schematic outlining the CRISPR cell generation as Fig. 3A and its description on line 926.

**Author response image 2. sa4fig2:** 

1. To clarify some of our previous (and persisting) concerns about Figure 3D/E - it is true that a reduction in 25% of cell size is dramatic. But (if we understand correctly) your claim is that a reduction in 22% (this is a guess, as the actual numbers are not supplies) is significantly less than 25%. Even if it is, statistically speaking, significant, what is the physiological relevance of this very slight effect? In this experiment, the N was quite large, and we wonder if the images in D are representative - it would be nice to label the data points in E to highlight which images you used.

We now mention the average cell area contraction measurements in the legend to Fig. 3F on line 935. We also tracked down the individual cells shown in Fig. 3E and they are now labelled as data points in blue in Fig. 3F. HeLa cell collapse is a simplified model of EPHB2 function and we do not know whether the difference between the behaviour of CTRLCRISPR and MYCBP2 CRISPR cells is physiologically significant and thus we prefer not to speculate on this.

1. Figure 3F and other stripe assays - In the end, it is your choice how to quantify. We believe that quantifying area of overlap is a more informative and objective measurement that might actually benefit your analyses. That said, if you do keep the quantification as it is now, you have to define the threshold of what you mean by "cell/s (or an axon in 7A, where it is even more complicated as are you eluding to primary, secondary, or even smaller branches) are RESIDING within the stripe". Is 1% overlap sufficient or do you need 10 or 50% overlap?

We now added this statement to the methods on line 745: “A cell was considered to be on an ephrin-B2 stripe when more than 50% of its nucleus was located on that stripe”. For chick explant stripe assay, when measuring the length of an axon on a stripe, we only measured the main axons originated from the explants.

For explant/stripe experiments in Fig. 7 AB, we now use the term “GFP-expressing neurite” rather than “branch”. This was already present in the results of the previous version, but the methods and legend needed to be brought up to date (lines 786 and 1008). We think that “branch” was a confusing term that was supposed to mean the same thing as “neurite” but came across as some indication of branching. We do not know whether the GFP+ neurites were primary or secondary extensions of explants, or in fact, whether some of them contained more than one axon. We also adjusted the method to reflect the fact that some stripes were used in conjunction with a single explant and added a reference to a previous study extensively using this method (Poliak et al., 2015) on line 778.

1. We still don't get the link to the lysosomal degradation. Your data suggests that in your cells EPHB2 is primarily degraded by the lysosomal pathway and not proteasome. Any statement about MYCBP2 is not strongly supported by the data, in our opinion - Unless you develop some statistical measurement that shows that the effect of BafA1 is statistically different in MYCBP2 cells than in control cells. Currently, this is not the case and the link is therefore not warranted in our opinion.

We generated a new version of Fig. 4K with average increase in EPHB2 levels in the presence of BafA1 and CoQ, compared to DMSO treated controls (see below). BafA1 and CoQ restored EPHB2 protein levels by 19% and 14% respectively in CtrlCRISPR cells, while the inhibitors restored EPHB2 protein levels by 40% and 35% respectively in MYCBP2 CRISPR cells.

**Author response image 3. sa4fig3:** 

For each of the 4 replicates, the increase in EPHB2 levels by BafA1 compared to DMSO is as followsAuthor response table 1:

These values are not significantly different between CtrlCRISPR cells versus MYCBP2 CRISPR cells (p = 0.08, student’s t test). Similarly for the CoQ experiment. We now temper our conclusion for this experiment: Although the difference in percentage increase between CTRLCRISPR cells and MYCBP2CRISPR cells is not significant, this trend raises the possibility that the loss of MYCBP2 promotes EPHB2 receptor degradation through the lysosomal pathway (line 319). We also adjusted the section title (line 306).

1. While the *C. elegans* part is now MUCH better explained - we are not sure we understand the additional insight. The fact that vab-1 and glo4 double mutants are additive as are vab1 and fsn1, suggest they act in parallel (if the mutants are NULL, and not if they are hypomorphs, if one wants to be accurate) - how this relates to your story is unclear. The vab1/rpm1 double mutant is still uninformative and incomplete. rpm1 phenotype is so severe that nothing would make it more severe. We read the Jin paper that the authors directed to - nothing makes the rpm1 phenotype more severe. Yes, some DOWNSTREAM elements make the rpm1 phenotype LESS severe - this is not something you were testing, to the best of our knowledge. Rather, you wanted to see if rpm1 mutant resulted in stabilization of vab1 and thus suppression of vab1 phenotype - we are just not sure the system is amenable to test (actually reject) your hypothesis that Vab1 is degraded by rpm1. Also, assuming we are talking about NULLs, the fact that the rpm1 phenotype is WAY stronger than the vab1 mutant, suggests that rpm1 functions via multiple routes, adding even more complexity to the system. Given these results, despite the much improved clarity, we are still not sure that the worm data adds new insight, rather than potentially confusing the reader.

We realise that the genetic interactions between vab-1 and the RPM-1/MYCBP2 signalling network are complicated. However, we insist on keeping the data for the sake of its availability for future studies and completeness. We also think it is important for readers and the community to see these data, even if the authors and reviewers are not entirely in agreement about the importance/interpretation of experimental outcomes. It is our hope that the community will examine the results and draw their own conclusions.

A few points of clarification:

The *C. elegans* experiments were designed to test genetically if the vertebrate interactions between EPHB2 and MYCBP2 and its signalling network are conserved. We studied two kinds of interactions: (1) between vab-1 and RPM-1/MYCBP2 downstream proteins (GLO-4 and FSN-1) and (2) between vab-1 and rpm-1. For these studies, we used null alleles for vab-1, glo-4 and fsn-1 which is now noted on lines 440, 453, 475 and 859. Our findings are consistent with the VAB-1 Ephrin receptor functioning in parallel to known RPM-1 binding proteins. This is further supported by new data: vab-1; fsn-1 double mutants showed enhanced incidence of axon overextension defects using a second transgenic background, zdIs5 (Pmec-4::GFP), to visualize axon termination (Fig. 8F).

This second transgenic background also allowed us to generate new data to address your concerns about phenotypic saturation in rpm-1 mutants. To do this, we used the zdIs5 (Pmec4::GFP) genetic background, in which axon termination defects are not saturated in rpm-1 mutants (Fig. 8F) because they can be enhanced by other mutants such as cdc-42 and unc-33 (Fig. 7C, D, in Borgen et al. Development 144, 4658–4672 (2017), PMID 29084805). In this new background, we found that vab-1 loss of function fails to enhance the incidence of severe “hook” defects in rpm-1 mutants which is an indication that the two genes function in the same pathway. Importantly, prior studies in this background, also showed that mutants in the RPM-1 signalling network (e.g. fsn-1, glo-4 and ppm-2) do not enhance the incidence of severe “hook” defects as double mutants with rpm-1 compared to rpm-1 single mutants (Fig. 7B, ibid.).

To reflect these ideas more clearly, we revised the Results section pertaining to *C. elegans* genetics (starting on line 418) and tempered our discussion (lines 517). Basically, this section now says that we studied genetic interactions between vab-1 and the RPM-1/MYCBP2 signalling network. From these experiments we conclude that: (1) The enhancement of overextension defects in vab-1; glo-4 and vab-1; fsn-1 double mutants compared to single mutants indicates that VAB-1/EPHR functions in parallel to known RPM-1 binding proteins to facilitate axon termination, and (2) Since the vab-1; rpm-1 double mutants do not display an increased frequency or severity of overextension defects compared to rpm-1 single mutants, VAB-1 /EPHR functions in the same genetic pathway as RPM-1/MYCBP2.

The new genetic data included in this version were generated by Karla J. Opperman who is now included as a co-author.

Further corrections:

**Author response image 4. sa4fig4:** 

Because of the errors associated with quantifications in Fig. 3D (see above), we reviewed other quantification methodologies and noticed another discrepancy that required a correction. In the hippocampal neuron growth cone collapse assay shown in the previous version of Fig. 7 D (left), the growth cones were classified into three groups: 1, fully collapsed; 2, hard to tell, but not fully collapsed; 3, fan-shape cones. Two different quantifications were performed as follows: (1), number of fully collapsed cones divided by the numbers of all growth cones; (2), number of fully collapsed cones divided by [number of fully collapsed cones + fan-shape cones]. CC erroneously used the second method to generate Fig. 7D.

We think that the first method is more appropriate. Furthermore, since n=5 for the Fc and eB1-Fc conditions, but n=3 for the eB2-Fc condition, we decided to omit it. The final plot for figure 7D is the following:

**Author response image 5. sa4fig5:** 

Our conclusion still stands that exogenous FBD1 WT overexpression impaired the growth cone collapse mediated by EphB.

**Author response table 2. sa4table1:** 

Ctrl^CRISPR^ cells	+9%	+24%	+27%	+15%
MYCBP2^CRISPR^ cells	+31%	+22%	+62%	+39%